



# Flash drought characteristics based on three identification methods in the North China Plain, China.

Siyao Zhang[a], Jianzhu Li[a], Ting Zhang[a], Ping Feng[a]

[a] State Key Laboratory of Hydraulic Engineering Intelligent Construction and Operation, Tianjin University, Tianjin 300350, China

*Correspondence to:* Jianzhu Li *(lijianzhu@tju.edu.cn)*

**Abstract.** Flash drought (FD) is an onset and intensify rapidly type of drought that can harm the terrestrial ecosystem, and cause economic and agricultural losses. The North China Plain (NCP) is an important agricultural region in China where sustainable development is restricted by the frequent droughts and insufficient water resources. Coping with FD requires an understanding of the FD onset and identification in the NCP. Based on root zone soil moisture (RZSM), standardized evaporative stress ratio (SESR) and multiples of mean evaporative stress ratio (MESR), this study identified the FD events in the NCP from 1981 to 2022, revealed the FD characteristics such as frequency, duration, severity and intensity, explored the temporal and spatial trend, determined the FD hotspots, and demonstrated the impact of FD identification thresholds on the FD identification. The frequency distributions of FD events identified by RZSM, SESR, and MESR are all high in the central and northern NCP and low in the southern, whereas the total duration is high in the southern and eastern NCP and low in the northern. As the FD intensity increases, the onset stage lengthens, the recovery stage shortens, the total duration reduces, and the severity declines. The FD affected areas from various FD identification methods exhibit significant and similar seasonal variations, primarily occurring from May to August. Besides, NCP is prone to extreme and exceptional FDs. The NCP has a decreasing tendency of the FD characteristics, and three hotspots with frequent and serious FD events are identified in the northwestern, eastern and southwestern NCP. The FD frequency is also significantly influenced by the thresholds in the identification methods. This study provides insights into the FD characteristics in the NCP, and clarifies its trend and hotspots, which may be valuable for FD understanding and adaptation.

**Key words.** Flash drought, duration, severity, intensity, hotspot.

## 1 Introduction

Regional long-term inadequate precipitation can result in persistent and slow drought disasters, which harm the ecology, agriculture, and economy (Li et al., 2020; Limones, 2021). The terrestrial water cycle accelerates and droughts becomes more frequent under global warming. Except for the long-term and slow droughts, some droughts that exhibit rapid occurrence and intensification also happen frequently, which are called flash drought (FD) (Deng et al., 2022; Yuan et al., 2023). FD is restricted by both water and energy, and its rapid onset complicates drought monitoring and forecasting (Yuan et al., 2023; Zhang et al., 2022a). The severe FD in the United States during the 2012 summer rapidly shifted from no drought to extreme drought in less than a month. Its rapid intensification made it difficult to capture, resulting in economic losses of more than $30 billion (Yuan et al., 2023). Examples of the FD events also include those in western Russia in 2010, southern Great Plains in 2015, and southern China in 2019 (Edris et al., 2023; Hunt et al., 2021; Wang and Yuan, 2021). Because of the severe impact of FD on society and environment, its occurrence and development have received much more attention worldwide (Deng et al., 2022; Li et al., 2020).

FD happens frequently accompanied with increasing temperature and decreasing precipitation, resulting in a rapid increase in evapotranspiration and reduction in soil moisture (Tyagi et al., 2022). As a result, some hydrological and meteorological variables were applied as key indicators to identify FD in previous studies. Currently, the FD identification methods can be mainly divided into three categories based on conventional drought indicators, soil moisture, and atmospheric evaporation



demand, respectively. The conventional drought indicators used for identifying FD include the United States Drought Monitor
(USDM) (Chen et al., 2019; Edris et al., 2023; Otkin et al., 2019; Pendergrass et al., 2020), Standardized Precipitation
Evapotranspiration Index (SPEI) (Fu and Wang, 2022; Noguera et al., 2020; Noguera et al., 2021), and Standardized
Precipitation Index (SPI) (Noguera et al., 2021; Parker et al., 2021). Chen et al. (2019) defined FD as a drought event with at
least two degrees of deterioration happening within four weeks using USDM, and they discovered that FD occurs mostly
during the warm season in the central United States. Otkin et al. (2019) also provided the FD evolution in the central and
southern United States from July to November 2015 using USDM. Noguera et al. (2020) used SPEI to identify the FD in Spain
from 1961 to 2018, illustrating that the northwestern Spain is more prone to FD. They also revealed a remarkable consistency
in the FD identified by SPI and SPEI (Noguera et al., 2021). However, the traditional drought indexes have drawbacks. SPI
only examines the regional precipitation deficit, neglecting the influence of climatic variables like temperature and evaporation.
SPEI describes the combined impacts of insufficient precipitation and increased evapotranspiration, yet its climatic response
is quite slow (Deng et al., 2022). Besides, USDM is only applicable in the United States and cannot be used in other regions.
As a result, several specific indicators for FD identification have also been created.

The influence of FD on agriculture cannot be disregarded. Soil moisture represents the combined influence of precipitation
and evapotranspiration. Hence, the soil moisture indicators have been frequently employed in the FD identification, such as
root zone soil moisture (RZSM, Yuan et al., 2019), soil moisture index (SMI, Hunt et al., 2009, 2021), soil moisture volatility
index (SMVI, Osman et al., 2021; Osman et al., 2022), and flash drought stress index (FDSI, Sehgal et al., 2021). Yuan et al.
(2019) investigated the rapid decreases of soil moisture in a short period, suggested an FD identification method based on
pentad RZSM, and examined the FD risk exposure in China. Hunt et al. (2021) utilized SMI to depict agricultural water stress
and illustrated how the FD affects soil moisture. Osman et al. (2021) developed SMVI to successfully capture FD in humid
and semi-humid regions of the United States, and explored its sensitivity. Sehgal et al. (2021) proposed FDSI, a combination
of relative rate of dry down (RRD) and soil moisture stress (SMS), to characterize FD and analyze its mechanism.

In addition to soil moisture, various studies have identified FD based on atmospheric evaporation demand. The evolutionary
stress index (ESI) indicates the normalized anomaly of the ratio of actual evapotranspiration (ET) to potential
evapotranspiration (PET) (Anderson et al., 2007; Hunt et al., 2021). Otkin et al. (2019) studied the vegetation stress in the
United States from 2001 to 2017 by ESI, and Hunt et al. (2021) also quantified the spatial evolution of the 2010 Russian FD
and its influence on agriculture. Ahmad et al. (2022) evaluated the occurrence and spread of two FD events in the northern
Great Plain in 2016 and 2017 using ESI as well. Li et al. (2020) developed an FD temporal and spatial tracking framework
based on standardized evapotranspiration deficit index (SEDI, Vicente-Serrano et al., 2010), and assessed the FD
characteristics and driving factors in the Pearl River Basin of China between 1960 and 2015. The evaporative demand drought
index (EDDI) detected FD through the PET response to surface dryness anomalies. Hobbins et al. (2016) contrasted USDM
and EDDI to depict the FD evolution in the United States. Hoffmann et al. (2021) identified FD under CMIP5 by SPI, EDDI,
and ESI. Christian et al. (2019) proposed a method for identifying FD based on standardized evaporative stress ratio (SESR),
which has been widely promoted and applied. Deng et al. (2022) used SESR to determine the global FD spatiotemporal
characteristics and its meteorological driving factors from 1981 to 2020. Zhong et al. (2022) utilized SESR to evaluate the FD
spatial and temporal characteristics in the Pearl River Basin of China, and Edris et al. (2023) applied it to quantify the FD
rapidly increasing components in the United States from 1979 to 2019.

RZSM and SESR are commonly utilized as FD indicators in regional FD study. Basara et al. (2019) measured the FD
spatiotemporal development and spread in the United States in 2012 by SESR, while Gou et al. (2022) used SESR to track the
FD path in the Huaibei Plain of China from 2001 to 2019. Zhang et al. (2022a) assessed the FD intensity based on RZSM in
China from 1979 to 2016, and examined the effectiveness of multiple linear regression (MLR), long short term memory
(LSTM), and random forest (RF) models in simulating the FD intensity. Mukherjee and Mishra (2022b) investigated the global
FD frequency and intensity, as well as their influencing factors using SESR and RZSM. Shah et al. (2022) employed RZSM



to determine the FD frequency variations and driving factors in Europe from 1950 to 2019.

However, the SESR application had some problems. When identifying FD in SESR, SESR and the change of SESR ($\Delta$SESR) theoretically follow normal distributions, therefore the 50th percentile ($SESR_{50th}$ and $\Delta SESR_{50th}$) equal 0, indicating that $\Delta SESR_{50th}$ denotes no change in SESR ($\Delta SESR = 0$), whereas $\Delta$SESR below the 40th percentile of $\Delta$SESR values ($\Delta SESR_{40th}$) denotes a declining in SESR. However, whether SESR and $\Delta$SESR follow a normal distribution remains to be determined, and if not, the $\Delta SESR_{40th}$ may not be less than 0. In the study by Gou et al. (2022), it was found that the 36th percentile of $\Delta$SESR ($\Delta SESR_{36th}$) corresponds to an increase in SESR, where $\Delta SESR_{36th}$ is greater than 0. This phenomenon may be because that SESR gradually decreases during periods without precipitation, and increases during the precipitation process. It is conceivable for $\Delta$SESR to be less than 0 during periods with no precipitation and larger than 0 during precipitation periods. Consequently, the FD events may be underestimated or overestimated. Furthermore, Christian et al. (2019) does not provide a criterion for SESR at the FD onset, but only a criterion about the minimum SESR at the FD onset stage greater than the 20th percentile ($SESR_{20th}$). This leads to the possibility that the SESR at the FD onset is below $SESR_{20th}$, even if all other criteria are satisfied. Therefore, it is necessary to add a criterion on the SESR at the FD onset.

A new method based on the multiples of the mean evaporative stress ratio (MESR) for FD identification has been developed to address the aforementioned problems in SESR identification method in this study. To compare the local climatology at different locations, MESR is used instead of SESR, without considering the probability density function (PDF) that evaporative stress ratio (ESR) follows. Furthermore, MESR and $\Delta$MESR are fitted by multiple PDFs and converted into percentiles. To guarantee that the thresholds of MESR and $\Delta$MESR are less than 0 with a lower level of MESR and a decrease in MESR, the variable thresholds are employed for identifying FD. The objectives of this study are to: (1) propose an improved FD identification method called MESR based on the SESR; (2) characterize the FD in the North China Plain (NCP) from 1981 to 2022 based on RZSM, SESR and MESR, and reveal the FD frequency, duration, severity and intensity; (3) investigate the FD temporal and spatial trend, identify the FD hotspots in the NCP, and analyze the impact of thresholds on the FD identification.

## 2 Materials and Methods

### 2.1 Study area

The NCP is located in 31 ~ 43 °N and 110 ~ 123 °E, and is the second biggest plain in China, as can be seen in Fig.1 (Wang et al, 2022; Yu and Deng, 2022). It borders the Bohai Sea and the Yellow Sea to the east, Taihang Mountain and Loess Plateau to the west, Yanshan Mountain to the north, and the Yellow River Basin to the south (Wu et al, 2022; Zhang et al, 2022b). The NCP spans around $3 \times 10^5$ km$^2$, and the elevation is 3 ~ 2836 m, with most regions below 50 m (Wu et al, 2022; Yang et al, 2023). The NCP experiences a warm temperate semi-humid monsoon climate (Liu et al, 2018), with cold and dry winters and hot and humid summers (Zhao et al, 2017). It also has sufficient sunshine (Wang et al, 2022), with an average annual temperature of 6 ~ 17 °C and precipitation of 500 ~ 900 mm (Wu et al, 2022; Yu and Deng, 2022). Precipitation varies significantly by season, with the most precipitation occurring from June to August (Liu et al, 2018; Wu et al, 2022). With its flat terrain and fertile soil, the NCP is an important agricultural region and cropland production base in China (Liu et al., 2018; Yu and Deng, 2022). Wheat and corn are the main crops, along with grains, soybeans, and potatoes grown (Yu and Deng, 2022). The frequent drought disasters and fragile ecological environment of the NCP have restricted the sustainable development (Liu et al., 2018; Wang et al., 2023).

### 2.2 Data

Developed by the European Centre for Medium-Range Weather Forecasts (ECWMF), the fifth-generation European Center for Medium-Range Weather Forecasts (ERA5)-Land has been produced by replaying the land component of the ECMWF ERA5 climate reanalysis, with improved nonlinear dynamical downscaling compared to ERA5





(https://cds.climate.copernicus.eu/cdsapp#!/dataset/reanalysis-era5-land; Muñoz Sabater et al., 2019; Deng et al., 2022). The ET, PET, and water volumes in soil layer 1 (0 ~ 7 cm), soil layer 2 (7 ~ 28 cm), and soil layer 3 (28 ~ 100 cm) from 1981 to

2022 were acquired with the 0.1°×0.1° spatial resolution and 1 hour temporal resolution. The soil moistures in soil layers 0 ~ 28 cm and 0 ~ 100 cm were determined using the weighted average with water volumes of the three soil layers. The soil moistures, ET and PET are utilized to calculate pentad mean RZSM, SESR, and MESR to identify the FD events.

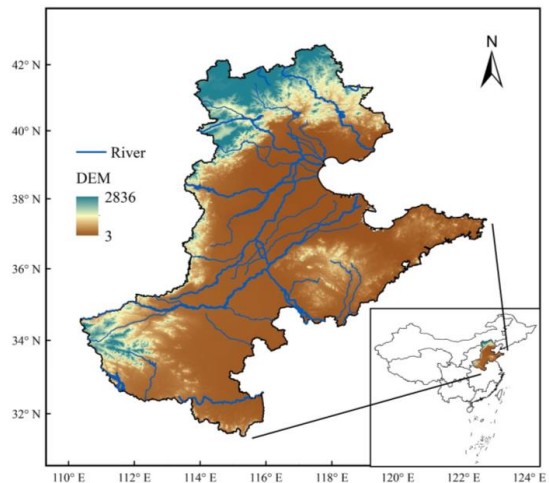

**Figure 1 Location and the digital elevation model (DEM) of NCP.**

**2.3 Methods**

**2.3.1 FD identification by RZSM (FD$_{RZSM}$)**

The FD identification by RZSM in this study was proposed by Yuan et al. (2019). Taking a grid as an example, the FD identification by RZSM is shown in Fig.2 (a). The steps are as follows.

(1) Convert the daily top 1-meter RZSM into percentile values (Chen et al., 2021). On a given day $p$ of the specified grid, a

sequence of RZSM from the $p$-1, $p$, and $p$+1 days of each year is created. Fit the RZSM sequence by the empirical distribution function (EDF) and then convert it into percentiles.

(2) Determine the pentad mean RZSM using the daily RZSM percentile values (Wang and Yuan, 2022).

(3) Identify FD$_{RZSM}$ by the following three criteria (Mukherjee and Mishra, 2022a):

a) The pentad mean RZSM decreases from above the 40th percentile (RZSM$_{40th}$) to below the 20th percentile (RZSM$_{20th}$)

within 4 pentads, on average decreasing by at least 5th percentile every pentad (for instance, August 1 ~ 11 in Fig.2 (a)).

b) The FD terminates when the decreasing RZSM recovers to above the RZSM$_{20th}$ for at least one pentad (e.g., August 31 in Fig.2 (a)). Additionally, the FD moves into the recovery stage when the average decline rate falls below 5th percentile or the RZSM percentile starts to increase (e.g., August 11 in Fig.2 (a)).

c) The FD lasts for at least 4 pentads and up to 18 pentads.

**2.3.2 FD identification by SESR (FD$_{SESR}$)**

Christian et al. (2019) proposed the FD identification method by SESR, which is based on ESR. On the basis of the method of Christian et al. (2019), a criterion regarding the SESR threshold at the FD$_{SESR}$ onset is added to the FD$_{SESR}$ identification in this study. An example of a FD event based on SESR is shown in Fig.2 (b).

(1) SESR calculation

ESR was calculated by taking the ratio between daily ET and PET (Christian et al., 2021):





$$ESR = \frac{ET}{PET} \qquad (1)$$

ESR was linearly detrended to eliminate the impacts of climate change on FD identification threshold over time. Then, the mean pentad ESR was calculated and standardized into SESR to remove the climate zone variations.

$$SESR = \frac{ESR - \overline{ESR}}{\sigma_{ESR}} \qquad (2)$$

where SESR is the Z score of ESR at the specific grid for a specific pentad, $\overline{ESR}$ and $\sigma_{ESR}$ are the mean and standard deviation of ESR at the specific grid for a specific pentad over the whole study period. Additionally, the SESR change is linearly detrended and normalized as:

$$(\Delta SESR)_Z = \frac{\Delta SESR - \overline{\Delta SESR}}{\sigma_{\Delta SESR}} \qquad (3)$$

where ΔSESR is the change in SESR between each pentad, $(\Delta SESR)_Z$ (referred to as ΔSESR) is the Z score of ΔSESR,

$\overline{\Delta SESR}$ and $\sigma_{\Delta SESR}$ are the mean and standard deviation of ΔSESR for an specific pentad at a given grid for all study period.

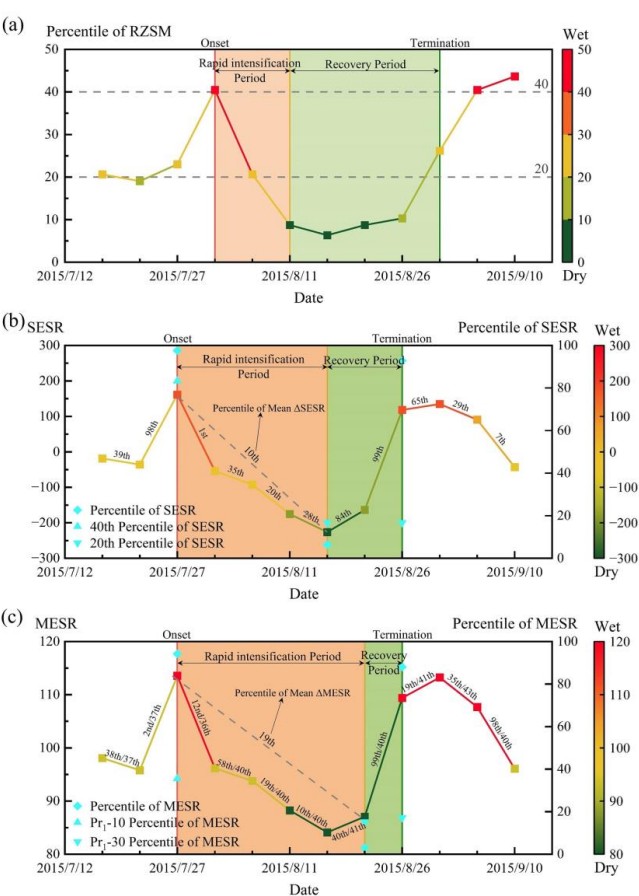

**Figure 2 Illustration of the definitions of FD through (a) RZSM, (b) SESR, and (c) MESR. Note that the numbers on the SESR line in (b) means the percentile of ΔSESR, the numbers before '/'on the MESR line in (c) means the percentile of ΔMESR, and that after '/' means Pr₂-10th.**

(2) FD$_{SESR}$ identification

Identify FD$_{SESR}$ by the following four criteria (Christian et al., 2019):

a) The minimum length is five pentads SESR changes, which means a length of six pentads.





b) The beginning SESR is above the 40th percentile of SESR values ($SESR_{40th}$), and the minimum SESR is less than the $SESR_{20th}$ (e.g., July 27 ~ August 16 in Fig.2 (b)). The FD event ends at the pentad when SESR exceeds $SESR_{20th}$.

c) No more than one consecutive ΔSESR is over the $ΔSESR_{40th}$. The ΔSESR after the ΔSESR exceeding $ΔSESR_{40th}$ must be lower than the $ΔSESR_{40th}$, and the ending SESR should be lower than the SESR two pentads before. The onset stage of FD starts from the first pentad of ΔSESR below the $ΔSESR_{40th}$, and ends at the last pentad with ΔSESR below the $ΔSESR_{40th}$.

d) The mean change in SESR during the FD event must be less than the 25th percentile of the ΔSESR for pentads that were encompassed with the FD events for all study period.

### 2.3.3 FD identification by MESR ($FD_{MESR}$)

The difference between MESR and SESR identification is that the MESR identification does not need to consider the PDF that ESR follows. In the meanwhile, the percentiles of MESR and ΔMESR for each pentad are fitted using the optimal PDF, and the variable thresholds that ensure less than 0 are employed in the process of FD identification. Figure 2 (c) illustrates a FD event based on MESR. The following are the steps of the FD identification by MESR:

(1) MESR calculation

Convert the mean pentad ESR that has been linearly detrended into $MESR_0$:

$$MESR_0 = \frac{ESR}{\overline{ESR}} \tag{4}$$

where $\overline{ESR}$ is the mean value of ESR, and $MESR_0$ is the multiple of $\overline{ESR}$ at the specific grid for a specific pentad. On the given pentad $p$ of the specified grid, a sequence of $MESR_0$ from the $p$-1, $p$, and $p$+1 pentads of each year are constructed.

Seven PDFs (normal distribution, generalized extreme value (GEV), chi-square, t distribution, logistic distribution, Pearson-III (P-III) distribution, and Rayleigh distribution) are used to fit the $MESR_0$ series. The optimal PDF ($f_1$) is determined with the lowest Akaike information criterion (AIC), Bayesian information criterion (BIC), and root mean square error (RMSE). Then the $MESR_0$ is converted into percentile values using $f_1$. Additionally, the $MESR_0$ change ($ΔMESR_0$) is detrended, and then converted into $ΔMESR_1$ by Eq. (5).

$$ΔMESR_1 = \frac{ΔMESR_0}{\overline{ΔMESR_0}} \tag{5}$$

where $ΔMESR_0$ is the change in MESR between adjacent pentads, $\overline{ΔMESR_0}$ is the mean value of $ΔMESR_0$, and $ΔMESR_1$ is the multiple of $\overline{ΔMESR_0}$ for the pentad for all study period. Similar as $MESR_0$, the $ΔMESR_1$ sequence is fitted by the seven PDFs, and $ΔMESR_1$ is converted into percentile values ($ΔMESR$) by the optimal PDF ($f_2$).

(2) Thresholds determination

The percentile of $MESR_0$ equals to 1 ($Pr_1$) and percentile of $ΔMESR_1$ equals to 0 ($Pr_2$) were calculated by the optimal PDF $f_1$ and $f_2$ at the specific grid for a specific pentad, as can be seen in Eqs. (6) and (7).

$$Pr_1 = \int_{-\infty}^{1} f_1(x)\,dx \tag{6}$$

$$Pr_2 = \int_{-\infty}^{0} f_2(x)\,dx \tag{7}$$

$Pr_1$ indicates that the ESR is at an average level and equals to the $\overline{ESR}$, whereas $Pr_2$ indicates that ESR remains unchanged

and the evaporative stress are unaltered as well. $ΔMESR_1$ from all the pentads during the FD onset stage for all years is fitted by the seven PDFs as well, and the optimal PDF is regarded as $f_3$. The percentiles of $ΔMESR_1$ from all the pentads during the FD onset stage for all years equals 0 ($Pr_3$) can be determined by Eq. (8).

$$Pr_3 = \int_{-\infty}^{0} f_3(x)\,dx \tag{8}$$

(3) FD identification

$FD_{SESR}$ can be identified by the following four criteria:





a) The minimum length is five pentad MESR changes, which means a length of six pentads.

b) During the onset stage, the beginning MESR is above the $Pr_1$-10th percentile of MESR values ($MESR_{Pr1-10th}$), and the minimum MESR is below the $Pr_1$-30th percentile ($MESR_{Pr1-30th}$) (e.g., July 27 ~ August 21 in Fig.2 (c)). The FD event ends at the pentad when MESR exceeds $MESR_{Pr1-30th}$.

c) No more than one consecutive ΔMESR is over the $Pr_2$-10th percentile of ΔMESR ($ΔMESR_{Pr2-10th}$). The ΔMESR after the ΔMESR exceeding $ΔMESR_{Pr2-10th}$ must be lower than the $ΔMESR_{Pr2-10th}$, and the ending MESR should be lower than the MESR two pentads before. The onset stage starts from the first pentad of ΔMESR below the $ΔMESR_{Pr2-10th}$, and ends at the last pentad with ΔMESR below the $ΔMESR_{Pr2-10th}$. The number

d) The mean change in MESR during the FD event must be less than the $Pr_3$-25th percentile of the ΔMESR and for pentads that were encompassed with the FD events for all study period.

### 2.3.4 FD characteristics

Seven FD characteristics are used in this study. The FD frequency is defined as the average number of FD events per year, and incidence rate (IR) as the percentage of years that the FD events occurred during the study period. Duration is measured as the pentads from the onset of the FD events to the termination ($duration_{Total}$), and is divided into the duration of onset and recovery stages ($duration_{Onset}$ and $duration_{Recovery}$). Severity of $FD_{RZSM}$ is determined by the accumulated deficit of RZSM relative to the 40th percentile during a FD event, that of $FD_{SESR}$ is determined by the accumulated deficit of SESR relative to the 40th percentile, and that of $FD_{MESR}$ is determined by the accumulated deficit of MESR relative to the $Pr_1$-10th percentile. Intensity is the mean change in RZSM, SESR and MESR percentiles during the FD onset stage. Table 1 shows that the FD events can be classified into four categories by intensity: moderate, severe, extreme and exceptional FDs (Christian et al., 2019; Mukherjee and Mishra, 2022b).

**Table 1 FD classification by intensity.**

| FD intensity index | FD classification | Intensity of RZSM, SESR and MESR |
| --- | --- | --- |
| FD1 | Moderate | > 20th |
| FD2 | Severe | 15 ~ 20th |
| FD3 | Extreme | 10 ~ 15th |
| FD4 | Exceptional | < 10th |

### 2.3.5 Mann-Kendall trend test

Without assuming a probable probability distribution, Mann-Kendall test is extensively used to determine the trend with consistently increasing or decreasing (Mann, 1945; Mahto and Mishra, 2023; Noguera et al., 2020). Compared with parametric tests, the nonparametric Mann-Kendall test is less likely to affect by outliers. The Mann-Kendall test is used to assess the trend of FD characteristics at the grid scale. The detailed calculation steps of Mann-Kendall test can be seen in Mann (1945).

### 2.3.6 Indicator to identify hotspots

Taking into account the FD characteristics, higher frequency and severity, as well as lower intensity, correspond to FD that may cause greater harm. Therefore, an indicator to identify FD hotspots is constructed based on frequency, severity and intensity of $FD_{RZSM}$, $FD_{SESR}$ and $FD_{MESR}$, and it can be calculated as follows:

$$\text{frequency}' = \frac{\text{frequency} - \text{frequency}_{min}}{\text{frequency}_{max} - \text{frequency}_{min}} \quad (9)$$

$$\text{severity}' = \frac{\text{severity} - \text{severity}_{min}}{\text{severity}_{max} - \text{severity}_{min}} \quad (10)$$

$$\text{intensity}' = \frac{\text{intensity} - \text{intensity}_{min}}{\text{intensity}_{max} - \text{intensity}_{min}} \quad (11)$$





$$\text{Hotspots} = \frac{1}{3}\left(\overline{\text{frequency'}} + \overline{\text{severity'}} + \frac{1}{\overline{\text{intensity'}}}\right) \qquad (12)$$

where frequency$_{max}$, frequency$_{min}$, severity$_{max}$, severity$_{min}$, intensity$_{max}$, and intensity$_{min}$ are the maximum and minimum
frequency, severity, and intensity in the NCP; frequency', severity', and intensity' are the max-min normalization of frequency,
severity, and intensity; and $\overline{\text{frequency'}}$, $\overline{\text{severity'}}$, and $\overline{\text{intensity'}}$ are the mean frequency, severity, and intensity of FD$_{RZSM}$,
FD$_{SESR}$ and FD$_{MESR}$.

## 3 Results

### 3.1 Shortcomings of FD$_{SESR}$ and rationality of FD$_{MESR}$

To graphically illustrate the limitations of the FD$_{SESR}$ method, the distribution of SESR$_{50th}$ and ΔSESR$_{50th}$ are shown in Fig.3
(a) (b). Theoretically, SESR and ΔSESR follow the normal distribution, where SESR$_{50th}$ = 0 and ΔSESR$_{50th}$ = 0, indicating that
SESR$_{40th}$ is below the average level and ΔSESR$_{40th}$ is less than 0 with the decreasing SESR. However, the skewed distributions
of SESR$_{50th}$ and ΔSESR$_{50th}$ in NCP demonstrate that SESR$_{50th}$ and ΔSESR$_{50th}$ are generally not 0. When SESR$_{50th}$ > 0, it is
possible that SESR$_{40th}$ > 0, indicating that ESR$_{40th}$ exceeds $\overline{ESR}$ and SESR$_{40th}$ cannot indicate the low evaporative stress. 
When ΔSESR$_{50th}$ > 0, maybe the corresponding ΔSESR$_{40th}$ > 0, reflecting an increasing SESR, which could result in the
underestimation of evaporative stress and inaccurate capture of FD events that would not occur. When SESR$_{50th}$ < 0, ESR$_{40th}$
may be significantly lower than $\overline{ESR}$, which would indicate a lower evaporative stress. When ΔSESR$_{50th}$ < 0, ΔSESR$_{40th}$ is
also significantly less than 0, indicating a severe decreasing SESR, leading to the overestimation of evaporative stress and
disregard for the FD events that actually occurred.

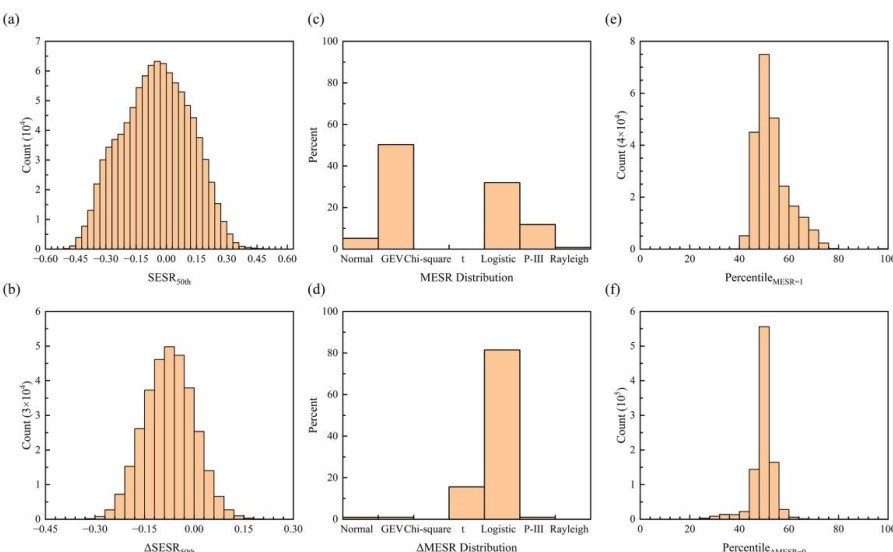

**Figure 3 The distribution of (a) SESR$_{50th}$, (b) ΔSESR$_{50th}$, the optimal PDF that (c) MESR$_{50th}$ and (d) ΔMESR$_{50th}$ follow, and distribution of (e) Pr$_1$ and (f) Pr$_2$.**

Fig.3 (c) (d) shows the optimal PDFs that MESR and ΔMESR follow ($f_1$ and $f_2$). The typically followed distributions for MESR
($f_1$) are mainly GEV and logistic distributions, and those for ΔMESR ($f_2$) are logistic and t distributions. Since MESR and
SESR are both obtained by the linear transformations to ESR, it can be assumed that MESR follows the skewed distribution
as well, corresponding to Fig.3 (a) (b). The Pr$_1$ and Pr$_2$ are shown in Fig.3 (e) (f). While there are also Pr$_1$ and Pr$_2$ larger or
smaller than the 50th percentile, there has been a notable improvement compared with SESR. Therefore, the FD$_{MESR}$ method,





which fits several PDFs and uses the variable thresholds for FD identification, may be employed to identify more accurate FD.

### 3.2 Spatial characteristics of FD events

RZSM, SESR and MESR were utilized to identify FD events between 1981 and 2022, and the spatial distribution of their characteristics were determined. Figure 4 depicts the spatial distribution patterns of frequency, duration$_{Total}$, severity and intensity. The frequency distribution and the variation with latitude are also demonstrated. Those of IR, duration$_{Onset}$ and duration$_{Recovery}$ can also be seen in Fig.S1.

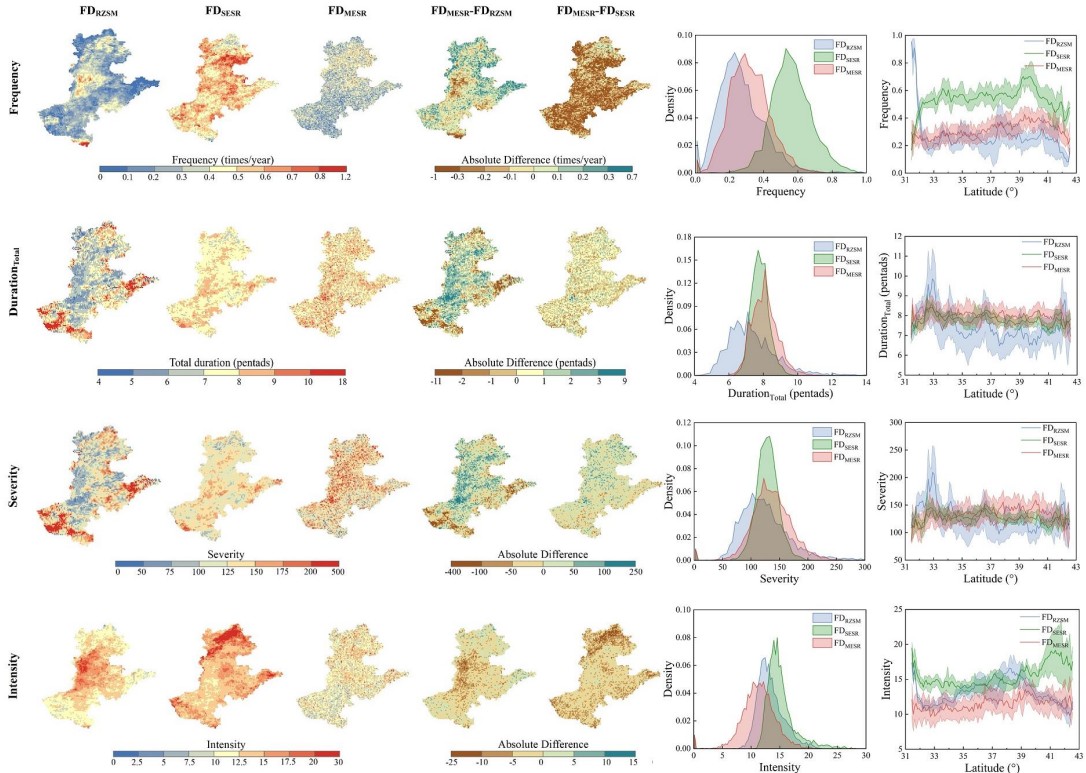

**Figure 4 Spatial distributions of the frequency, duration, severity and intensity of FD events identified by RZSM, SESR and MESR, and their absolute differences, as well as the frequency distribution of the characteristics and its variation with latitude.**

The frequency of FD$_{RZSM}$ is high in the central and northern NCP and low in the southern NCP with two high-frequency regions in the central and northern NCP, that of FD$_{SESR}$ is high in the central and northeastern NCP and low in the southern NCP, and that of FD$_{MESR}$ is higher in the northern NCP than the southern. The lack of precipitation and increase of evapotranspiration would hasten the decrease of soil moisture and increase the likelihood of FD in the northern NCP than the southern NCP (Gou et al., 2022; Yuan et al., 2023). In the northern NCP, evapotranspiration is water limited, and the inadequate precipitation is the primary driving factor of FD. The latitude distribution of frequency also verifies this. The FD$_{RZSM}$ frequency decreases sharply within the range of 31 ~ 33 °N, and then fluctuates steadily in the north of 33 °N. In the region north of 33 °N, the FD$_{RZSM}$ frequency likewise peaks around 38 °N and 41 °N, respectively, and slowly decreases in regions north of 41 °N. The FD$_{SESR}$ and FD$_{MESR}$ frequency increases with latitude in the region south of 40 °N, reaches the maximum at 40 °N, and then gradually decreases in the region north of 40 °N. Furthermore, the overall FD$_{SESR}$ frequency is higher than that of FD$_{RZSM}$ and FD$_{MESR}$. The density distribution demonstrates that the frequency of FD$_{RZSM}$ and FD$_{MESR}$ are both mostly dispersed 0 ~ 0.6 times/year, whereas the FD$_{SESR}$ frequency is larger than that of FD$_{RZSM}$ and FD$_{MESR}$, ranging between 0.4 and 0.8 times/year.





The difference in frequency spatial distribution reveals that the frequency difference between $FD_{MESR}$ and $FD_{RZSM}$ is between -0.2 and 0.3 times/year, with $FD_{MESR}$ has a higher frequency in the northern and eastern NCP, and a lower frequency in the central and southern NCP. With the exception of northeastern corner of NCP where $FD_{MESR}$ frequency is $0 \sim 0.2$ times/year higher than that of $FD_{SESR}$, the $FD_{MESR}$ frequency is mainly $0.1 \sim 0.3$ times/year lower in the other region of NCP. The IR spatial distributions for the three FD methods are fairly similar to those of frequency (Fig.S1). Figure S2 compares the characteristics of $FD_{RZSM}$, $FD_{SESR}$, and $FD_{MESR}$. Figure S2 (a) (b) depict the frequency and IR scatter distributions, which reveal a more significant linear relationship between $FD_{RZSM}$ and $FD_{MESR}$ than that between $FD_{RZSM}$ and $FD_{SESR}$. Besides, the frequency and IR scatter distributions of $FD_{SESR}$ and $FD_{MESR}$ are quite comparable.

The duration$_{Total}$ distributions of $FD_{RZSM}$, $FD_{SESR}$, and $FD_{MESR}$ differ, but they are all high in the southern and eastern NCP and low in the northern NCP. The duration$_{Total}$ of $FD_{RZSM}$ in the southwestern and eastern NCP surpasses 10 pentads, whereas that of $FD_{SESR}$ in the all NCP reaches $7 \sim 10$ pentads. The latitude distribution demonstrates that the $FD_{RZSM}$ duration$_{Total}$ shows great fluctuations with latitude, peaking at 33 °N and then gradually decreasing north of 33 °N. The duration$_{Total}$ of $FD_{SESR}$ and $FD_{MESR}$ varies similarly with latitude, with greatest duration$_{Total}$ at 33 °N, and gradually decreasing north of 33 °N. Warmer temperatures may result in longer FD duration (Zhang et al., 2022c). The density distribution also reveals a considerable similarity in the duration$_{Total}$ distribution between $FD_{SESR}$ and $FD_{MESR}$, which is predominantly distributed between 7 and 9 pentads, but that of $FD_{RZSM}$ is relatively low, distributed between 5 and 9 pentads. This is connected to their identification methods. The differences in duration$_{Total}$ indicate that the $FD_{MESR}$ duration$_{Total}$ is lower than that of $FD_{RZSM}$ in the southwestern and eastern NCP. While the duration$_{Total}$ difference of $FD_{MESR}$ and $FD_{SESR}$, which ranges from -1 to 1 pentads, is not much significantly different in the NCP. Both $FD_{RZSM}$ and $FD_{SESR}$ have duration$_{Onset}$ that are mostly 2 to 4 pentads, with $FD_{SESR}$ having a greater duration$_{Onset}$ than $FD_{RZSM}$. The $FD_{MESR}$ duration$_{Onset}$, with $2 \sim 5$ pentads, is greater than that of the $FD_{RZSM}$ and $FD_{SESR}$. Besides, duration$_{Recovery}$ for $FD_{RZSM}$ is $4 \sim 6$ pentads, but for $FD_{SESR}$ and $FD_{MESR}$ are both ranges from 2 to 7 pentads. Similar to duration$_{Total}$, the duration$_{Onset}$ spatial distributions display a pattern of high in the southern and low in the northern NCP, as well as the duration$_{Recovery}$ for $FD_{RZSM}$ (Fig.S1). Besides, the duration$_{Recovery}$ for $FD_{SESR}$ is mainly $4 \sim 5$ pentads, and that of $FD_{MESR}$ is low in the northern NCP and high in the northern.

The $FD_{RZSM}$ severity is high in southwestern and eastern NCP and low in the central and northern NCP, while $FD_{SESR}$ severity is greater in the south and lower in the north. Whereas, $FD_{MESR}$ has high severity in the northern and central NCP and low severity in the southern NCP. Warming not only lengthens the drought durations, but it also exacerbates them by increasing surface evapotranspiration losses and decreasing the soil moistures (Yuan et al., 2019; Zhang et al., 2021). The $FD_{RZSM}$ severity peaks at 33 °N and then has a notable decrease as latitude increases. On the other hand, there is a stable distribution in the $FD_{SESR}$ severity. The $FD_{MESR}$ severity is relatively high in the $37 \sim 40$ °N. The severity distributions show notable differences. The $FD_{RZSM}$ and $FD_{MESR}$ severity is mostly concentrated between 50 and 200, whereas $FD_{SESR}$ severity ranges between 80 and 180. However, the severity distribution of $FD_{SESR}$ is sharpest and thinnest, whereas that of $FD_{RZSM}$ is shortest and fattest. The severity spatial distribution differences indicate that the $FD_{RZSM}$ severity is greater than the $FD_{MESR}$ severity in the southwestern and eastern NCP, but the $FD_{MESR}$ severity is greater in other regions. While the severity difference between $FD_{MESR}$ and $FD_{SESR}$ in the NCP is mostly between -50 and 50. In general, Fig.S2 (f) shows a better linear relationship between $FD_{RZSM}$ and $FD_{MESR}$ than that between $FD_{RZSM}$ and $FD_{SESR}$.

The lower the intensity, the more severe the FD event. The southern, northern and eastern NCP have smaller $FD_{RZSM}$ intensity values, while the central NCP has larger. The intensity value of $FD_{SESR}$ is the largest, and its spatial distribution is smaller in the central NCP and larger in the northern and eastern NCP. Whereas $FD_{MESR}$ has larger intensity value in the north of the NCP and lower in the south. The $FD_{RZSM}$ intensity value subsequently increases with the increasing latitude in the region south of 38 °N, reaching its maximum at 38 °N, and then gradually decreases in the region north. The $FD_{SESR}$ intensity increases with latitude in the region south of 41 °N, and then gradually decreases in the region north of 41 °N. Besides, the $FD_{MESR}$ intensity gradually increases with latitude and peaks at 39 °N, and then decreases in the region north of 39 °N. There are notable



differences in the intensity distribution among the three methods. The FD$_{RZSM}$ and FD$_{SESR}$ intensity is primarily concentrated

between 10 and 20, but that of FD$_{MESR}$ is between 5 and 20. The intensity spatial distribution difference between FD$_{RZSM}$ and FD$_{MESR}$ indicates that intensity of FD$_{MESR}$ is larger in the northern NCP but smaller in the central NCP, and the difference is relatively small, mainly between -10 and 5. Besides, the FD$_{MESR}$ intensity in the NCP is smaller than that of FD$_{RZSM}$ in the NCP, especially in the northern and southern NCP. In Fig.S2 (g), the FD$_{MESR}$ intensity is wider than FD$_{SESR}$ intensity, which is concentrated and ranges much smaller. Besides, the intensity between FD$_{RZSM}$ and FD$_{MESR}$ are more linear correlation than that

between FD$_{RZSM}$ and FD$_{SESR}$.

The FD characteristics of FD1 ~ 4, such as duration$_{Onset}$, duration$_{Recovery}$, duration$_{Total}$, and severity, were explored. Fig.5 illustrates their distributions of FD1 ~ 4 based on RZSM, SESR and MESR, and the main ranges are shown in Table S1. For the three FD identification methods, the duration$_{Onset}$ gradually lengthens with the increasing intensity grades, whilst the duration$_{Recovery}$ and duration$_{Total}$ shorten. That is, the higher the intensity grades of the FD event, the longer the onset stage, the

shorter the recovery stage and the total duration. That may be because that the intensity is derived from the RZSM, SESR and MESR change during the onset stage. The higher the intensity grades, the smaller the intensity values, and the longer onset stage, thereby shortening the recovery stage. Besides, the greater FD grades have the lower severity.

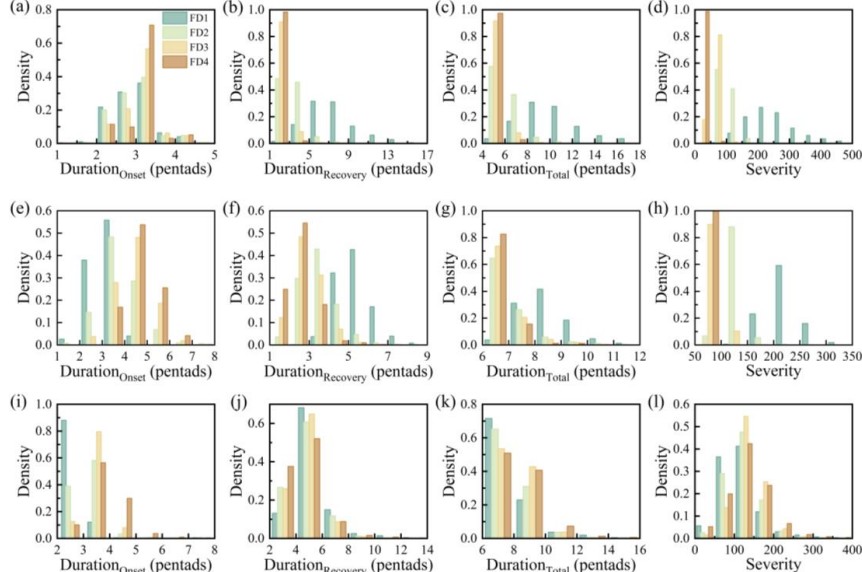

**Figure 5 Frequency distribution of the properties under different (a) ~ (d) FD$_{RZSM}$, (e) ~ (h) FD$_{SESR}$, and (i) ~ (l) FD$_{MESR}$**
**grades. The density means the proportion of grids with corresponding characteristics, such as for FD$_{RZSM}$1 events, the proportion of grids with severity between 200 and 250 to all grids is 0.27.**

Figure 6 depicts the average monthly affected area of FD events from 1981 to 2022. The affected area of FD$_{RZSM}$ events has evident seasonal and comparable seasonal fluctuations, with the majority concentrated from May to August. Almost 40% and 30% of the FD$_{RZSM}$ events occurred in June and July, more than 10% in May, almost 10% in August, and more than 5% in

April in the NCP. The affected area of FD$_{SESR}$ events is not as evident as that of FD$_{RZSM}$, which is concentrated from June to August. The affected area of FD$_{SESR}$ from June to August are all over 10%, and that in other months expect for January are almost 10%. Whereas, there is no seasonal fluctuations in the affected area of FD$_{SESR}$, where the affected area of FD$_{SESR}$ is about 10%.

Figure S3 shows the amounts of FD1 ~ 4 grids from January to December. The FD$_{RZSM}$ and FD$_{SESR}$ grids under different

intensity grades vary greatly with the season, while the seasonal fluctuations of FD$_{MESR}$ are small. FD$_{RZSM}$ occurs more frequently from May to August, while FD$_{SESR}$ from March to October. FD$_{RZSM}$ is mainly focused on FD$_{RZSM}$3, with FD$_{RZSM}$1





and $FD_{RZSM}2$ mainly concentrated in June and July, while $FD_{RZSM}4$ occurs more stably from April to August. For $FD_{SESR}$, $FD_{SESR}3$ also accounts for a large proportion, followed by $FD_{SESR}2$ and $FD_{SESR}4$, while $FD_{SESR}1$ occurs steadily throughout one year. $FD_{MESR}4$ in $FD_{MESR}$ occurred significantly, and the grid amounts of $FD_{MESR}3 \sim 4$ far exceeded that of $FD_{MESR}1 \sim 2$.

On the whole, the findings of the three FD identification methods show that the proportion of FD3 ~ 4 in the NCP is high, that is, NCP is prone to extreme and exceptional FDs.

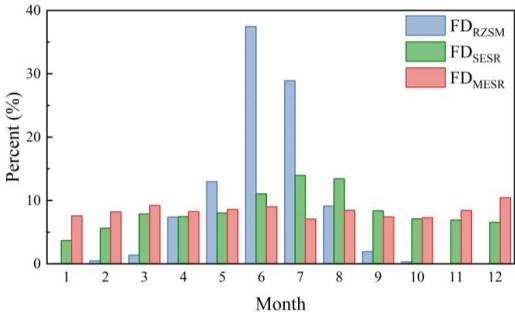

**Figure 6 Monthly average FD affected area percentages (%) identified by RZSM, SESR, and MESR from 1981 to 2022.**

### 3.3 Annual and spatial trend of FD characteristics

Figure 7 (a) depicts the annual affected area of FD events from 1981 ~ 2022. The affected areas of $FD_{RZSM}$ fluctuate between 0 and 80% with a clear decrease trend, while that of $FD_{SESR}$ and $FD_{MESR}$ ranging from 0 to 40% gradually rises. Figure 7 (b) ~ (f) show the annual variations in $FD_{RZSM}$, $FD_{SESR}$ and $FD_{MESR}$ characteristics as well. $FD_{MESR}$ has a longer $duration_{Onset}$ (3 ~ 4 pentads) compared to $FD_{RZSM}$ and $FD_{SESR}$ (2.5 ~ 3.5 pentads). Except for the quiet slightly decreasing trend of $FD_{RZSM}$ $duration_{Onset}$, both $FD_{SESR}$ and $FD_{MESR}$ $duration_{Onset}$ are decreasing with a very slight trend. Besides, the $duration_{Recovery}$ of

$FD_{RZSM}$, $FD_{SESR}$ and $FD_{MESR}$ all decreased steadily and have similar fluctuations, primarily ranges from 4 to 6 pentads. The $duration_{Total}$ of $FD_{SESR}$ and $FD_{MESR}$ gradually decreases within 7 ~ 9 pentads, whereas $FD_{RZSM}$ $duration_{Total}$ ranging from 5 to 10 pentads with severe oscillations displays a larger decline tendency than that of $FD_{SESR}$ and $FD_{MESR}$. This may be related to the set minimum duration in the FD identification. $FD_{RZSM}$, $FD_{SESR}$ and $FD_{MESR}$ have similar severity, and all are significantly reducing, with $FD_{RZSM}$ showing a more severe decrease tendency. The intensity values of $FD_{RZSM}$ significantly increases, and the increase in $FD_{SESR}$ is smaller than that of $FD_{RZSM}$. Whereas, the intensity values of $FD_{MESR}$ shows a gradually decreasing

trend. Overall, the durations and severity of the three FD identification methods are similar, with the decreasing tendency. As a result, the FD events in the NCP are being steadily reduced.

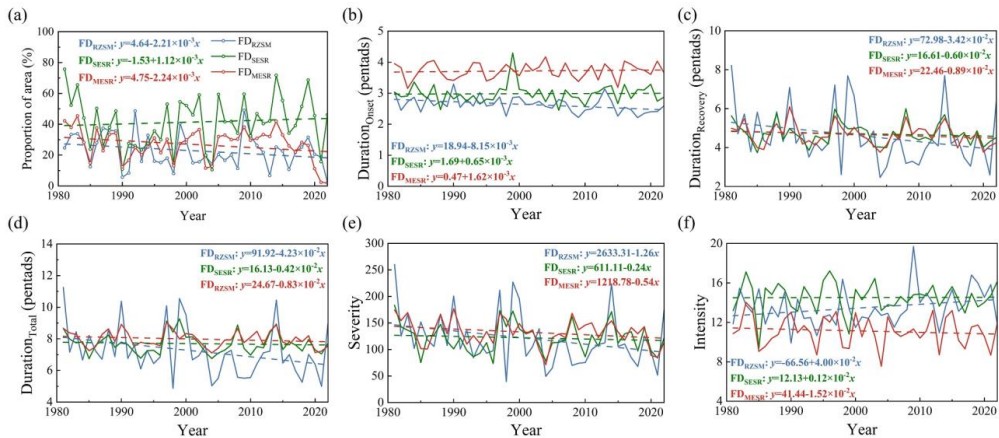

**Figure 7 Time series of (a) affected area percentages (%), (b) $duration_{Onset}$, (c) $duration_{Recovery}$, (d) $duration_{Total}$, (e)**
**severity, and (f) intensity of FD events over NCP from 1981 to 2022, with their linear trend.**





The Mann-Kendall trend test is used to examine the ratio of time when FD events occur, duration$_{Total}$, severity, and intensity of FD on the grid, as illustrated in Fig.8. Overall, the FD ratio, duration$_{Total}$, severity and intensity in the NCP are decreasing except for intensity of FD$_{RZSM}$ and FD ratio of FD$_{SESR}$, and their spatial distribution trends are comparable. Among them, the FD$_{RZSM}$ ratio in the southwestern NCP decreased significantly ($\alpha < 0.05$), and the FD$_{SESR}$ ratio in the southern and eastern NCP shows an increasing trend. The FD$_{MESR}$ ratio significantly decreases in the central NCP as well. The duration$_{Total}$ and severity of FD$_{RZSM}$ have all reduced. Whereas the intensity of FD$_{RZSM}$ increases in the central and northern NCP. Aside from that, the FD$_{SESR}$ duration$_{Total}$ and severity increases sporadically, while it tends to decrease as a whole. The FD$_{SESR}$ intensity value decreases in the northern NCP but increases in the southern. Although the dispersed northeastern NCP shows a significant decreasing trend, the duration$_{Total}$ and severity of FD$_{MESR}$ in the southern NCP increases, and the intensity decreases. The trend spatial distribution of FD characteristics in NCP is consistent with their annual trends, and are a decreasing tendency overall.

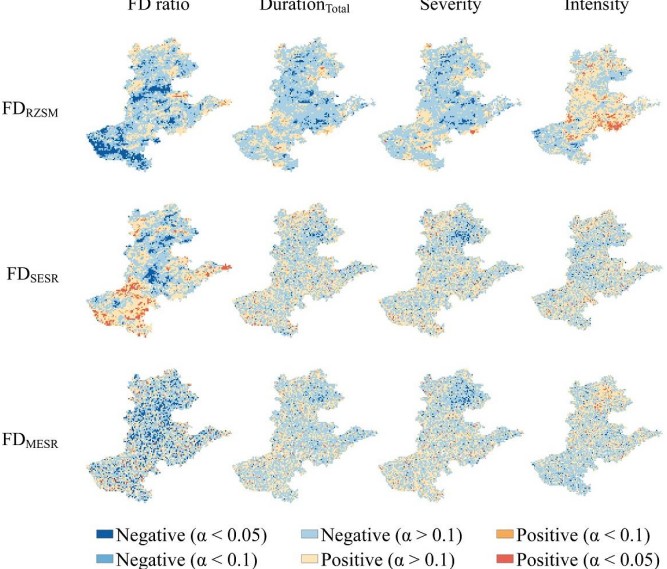

**Figure 8 Spatial trends of FD identified by RZSM, SESR, and MESR during 1981 ~ 2022.**

### 3.4 Typical historical events

To demonstrate the applicability of three FD identification methods, an evaluation is conducted based on the historical records of typical drought events. According to the records of "Flood and drought disasters in the Haihe River Basin" (Haihe River Water Conservancy Commission, MWR, 2009), the great summer drought occurred in Zhangjiakou in 1981, Shijiazhuang in 1983, and Dezhou and Binzhou in 1989 were extracted. Due to the lack of more detailed records on the drought occurrence time, the number of pentads that FD events happened in the summer (from June to August) in 1981, 1983 and 1989 were calculated, as shown in Fig.9.

Regarding the drought in Zhangjiakou in 1981, FD$_{RZSM}$ did not identified its occurrence, FD$_{SESR}$ identified that the drought occurred in the southeast and northeast of Zhangjiakou, and FD$_{MESR}$ identified a more dispersed and severe drought than FD$_{SESR}$. For the drought in 1983, FD$_{RZSM}$ identified the severe drought in the east, and its findings were more reasonable than that of the other two methods, where almost no drought was detected. The drought identified by FD$_{RZSM}$ in 1989 mainly occurred in Dezhou and the western Binzhou, and it has a shorter drought duration than FD$_{SESR}$. The FD$_{SESR}$ identified severe drought in southern Dezhou and Binzhou, with a long duration. Compared with the findings of FD$_{RZSM}$ and FD$_{SESR}$, the regions encountering FD$_{MESR}$ are more dispersed, and the drought duration is similar to that of FD$_{RZSM}$. Therefore, FD$_{MESR}$, FD$_{RZSM}$, and FD$_{SESR}$ have shown better performance in the drought identification in 1981, 1983 and 1989, respectively.





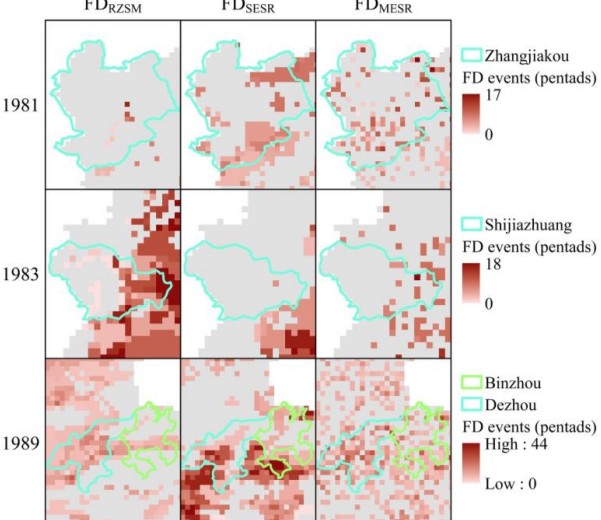

**Figure 9 Three FD events occurred in the summer of 1981, 1983, and 1989 in the Cities Zhangjiakou, Shijiazhuang, and Binzhou & Dezhou, respectively. The color bands represent the pentads with FD events from June to August of that year.**

### 3.5 Hotspots of FD in the NCP

As shown in Sect.3.4, the three methods for identifying FD may perform well in different events. Therefore, the hotspot identification indicator is constructed based on the frequency, severity and intensity of the three identification methods simultaneously. The hotspot identification indicator percentile is illustrated in Fig.10. Three hotspots with frequent and severe FD events in the NCP, where hotspot indicator percentile is above 60th, are detected. These hotspots are situated in the northwestern, eastern and southwestern NCP, respectively. High hotspot indicator regions are likely to encounter frequent and severe FD events. This might be because the regions are mostly with developed agricultural planting, high population and diversified human activities, increasing the moisture demand. The high-water demand causes the water scarcity, but the developed agriculture increases ET and the usage of soil moisture, rendering it prone to FD events. The NCP is one of the hotspots of FD worldwide (Christian et al., 2023).

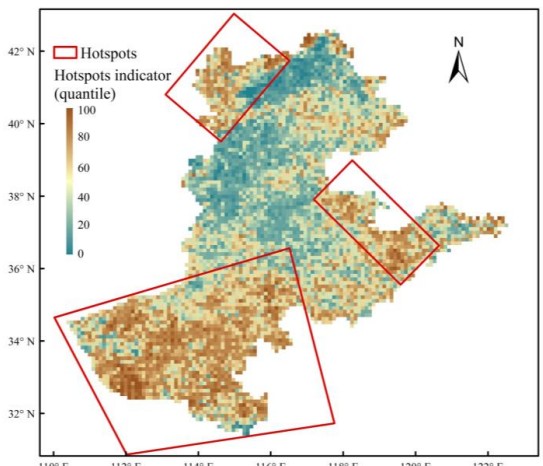

**Figure 10 Distribution of the FD hotspot identification indicator, as well as the four FD hotspots.**





### 3.6 Effect of all FD identification thresholds on the FD frequency

The frequency spatial distribution of FD identified by RZSM from various soil layer depths are presented in Fig.S4. When comparing the RZSM at the depths of $0 \sim 7$, $7 \sim 28$, $0 \sim 28$, and $28 \sim 100$ cm, as well as that of $0 \sim 100$ cm, it is evident that the frequency decreases with the soil depth. This might be as a result of that the deeper soil layers have less soil moisture variations and less impact from weather, but the surface soil layers react swiftly to the ET increasing and precipitation deficiency, which can capture more FD events. However, soil moisture in the deep soil layers promote and influence vegetation

growth directly. Therefore, identifying FD using soil moisture in the root zone makes more sense (Qing et al., 2022).

Based on the FD identification methods (see Sect.2.3.1, Sect.2.3.2 and Sect.2.3.3), the parameters in the three methods are summarized in Table 2, as well as their meanings and standard thresholds. Figs. 11, S7, and S8 display the FD frequency under different thresholds. Fig.11 demonstrates that the frequency under different 'RZSMpentad1' differ not much, suggesting that 'RZSMpentad1' is not sensitive to $FD_{RZSM}$ identification. However, their frequency is significantly influenced by other $FD_{RZSM}$

thresholds. Smaller 'RZSMonset1', 'RZSMpentad2' and 'MinDuration', and greater 'RZSMonset2' and 'RZSMtermination' are correlated with a greater $FD_{RZSM}$ frequency. As can be seen in Fig.S5, both 'SESRonset1' and 'ΔSESRonset' are frequency insensitive for $FD_{SESR}$, but smaller 'MinDuration' and 'SESRpentad', as well as greater 'SESRonset2' and 'MaxSESRchange', have greater $FD_{SESR}$ frequency. However, both the 'MESRonset1' and ΔMESRonset' in $FD_{MESR}$ are more sensitive to frequency than 'SESRonset1' and 'ΔSESRonset' in $FD_{SESR}$, with lower 'MESRonset1' and greater 'ΔMESRonset' correlating to higher

frequency, as illustrated in Fig.S6. Moreover, the effects of 'MinDuration', 'MESRonset2', 'MESRpentad', and 'MaxMESRchange' on $FD_{MESR}$ frequency are comparable to those of $FD_{SESR}$.

**Table 2 Parameters in the FD identification methods based on RZSM, SESR and MESR, as well as their meanings and standard thresholds.**

| FD identification method | Parameter | Meaning | Standard threshold |
|---|---|---|---|
| $FD_{RZSM}$ | RZSMonset1 | RZSM percentile that should exceed at the FD start | 40 |
| | RZSMonset2 | RZSM percentile that should decrease to during the FD onset stage | 20 |
| | RZSMpentad1 | Maximum pentads for RZSM to decrease from 'RZSMonset1' to 'RZSMonset2' | 4 |
| | RZSMtermination | RZSM percentile that should exceed at the FD termination | 20 |
| | RZSMpentad2 | Continuous pentads that RZSM should exceed the 'RZSMtermination' at the FD termination | 1 |
| | MinDuration | Minimum FD duration | 4 |
| $FD_{SESR}$ | MinDuration | Minimum FD duration | 6 |
| | SESRonset1 | SESR percentile that should exceed at the FD onset | 40 |
| | SESRonset2 | SESR percentile that should decrease to during the FD onset stage | 20 |
| | ΔSESRonset | ΔSESR percentile that should exceed at the FD onset stage | 40 |
| | SESRpentad | Continuous pentads that ΔSESR should not exceed the 'ΔSESRonset' at the FD onset stage | 1 |
| | MaxSESRchange | Percentile of the mean change of SESR during the entire FD event that should be below | 25 |
| $FD_{MESR}$ | MinDuration | Minimum FD duration | 6 |
| | MESRonset1 | MESR percentile that should exceed at the FD onset | $Pr_1$-10 |
| | MESRonset2 | MESR percentile that should decrease to during the FD onset stage | $Pr_1$-30 |
| | ΔMESRonset | ΔMESR percentile that should exceed at the FD onset stage | $Pr_2$-10 |
| | MESRpentad | Continuous pentads that ΔMESR should not exceed the 'ΔMESRonset' at the FD onset stage | 1 |
| | MaxMESRchange | Percentile of the mean change of MESR during the entire FD event that should be below | $Pr_3$-25 |





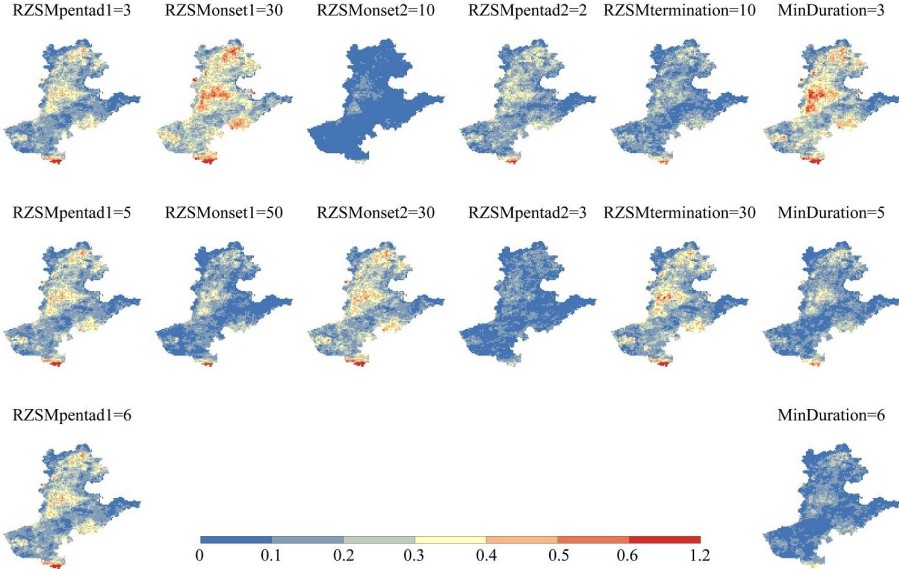

**Figure 11 Frequency for FD$_{RZSM}$ with different thresholds for the definition.**

## 4 Discussion

### 4.1 FD characteristics compared with previous studies

Based on RZSM, SESR and MESR, this study identified the FD events in the NCP from 1981 to 2022, and investigated the spatial distribution of their frequency, duration, severity and intensity. The findings suggest that the FD frequency in the NCP

is high in the north and low in the south, and the FD$_{RZSM}$, FD$_{SESR}$, and FD$_{MESR}$ frequency range mostly in 0 ~ 0.6, 0.4 ~ 0.8, and 0 ~ 0.6 times/year, respectively, while FD$_{RZSM}$, FD$_{SESR}$, and FD$_{MESR}$ IR range primarily from 10 to 50 %, from 30 to 60%, and from 10 to 50%, respectively. Previous studies have also disclosed the FD frequency. Mukherjee and Mishra (2022a) discovered the NCP suffering 3 to 12 FD$_{RZSM}$ events from 1980 to 2018. Yuan et al. (2019) estimated the FD$_{RZSM}$ frequency of about 0 ~ 6 times/decade in the NCP, and Wang and Yuan (2022) also identified the frequency of 2 ~ 6 times/decade between

1979 and 2020. Mukherjee and Mishra (2022b) utilized ERA5 to detect the FD$_{RZSM}$ IR in the NCP as 15 ~ 30 %, while that of Mahto and Mishra (2023) was 11 ~ 15 %. Gou et al. (2022) discovered that the FD$_{SESR}$ frequency in the Huaibei Plain, located south of the NCP, where suffered about 18 ~ 20 FD$_{SESR}$ events from 2001 to 2019. Deng et al. (2022) found that the FD$_{SESR}$ IR fluctuated from 25 % to 40 % in the NCP from 1981 to 2020, while Christian et al. (2023) simulated the global FD$_{SESR}$ events using the Coupled Model Intercomparison Project Phase 6 (CMIP6) models from 1980 to 2014 and found that IR in the NCP

was 25 ~ 35%. Considering the influences of data sources and spatial resolution, the FD frequency ranges in this study are comparable to those in previous studies, and the FD identification in this study appears to be reasonable.

This study distinguished the onset and recovery stages of FD events and identified their durations. The duration$_{Onset}$, duration$_{Recovery}$, and duration$_{Total}$ of FD$_{RZSM}$ are 2 ~ 3, 2 ~ 7, and 5 ~ 10 pentads, respectively, whereas those of FD$_{SESR}$ are 2 ~ 4, 4 ~ 6, and 7 ~ 10 pentads, respectively, and those of FD$_{MESR}$ are 3 ~ 5, 3 ~ 6, and 7 ~ 10 pentads, respectively. Yuan et al.

(2023) detected the worldwide FD$_{RZSM}$ events from 1951 to 2014, with the duration$_{Total}$ of around 30 ~ 40 days in the NCP; Zhang et al. (2020) discovered that the FD$_{RZSM}$ duration$_{Onset}$, duration$_{Recovery}$, and duration$_{Total}$ from 2003 to 2018 were 15 ~ 20, 5 ~ 30, and 25 ~ 50 days, respectively. Yuan et al. (2019) illustrated the average duration$_{Total}$ of 20 ~ 40 days from 1961 to 2005. Some studies also identified FD$_{SESR}$ events. Deng et al. (2022) revealed the FD$_{SESR}$ duration$_{Total}$ of 6 ~ 9 pentads in the NCP, and Gou et al. (2022) found the FD$_{SESR}$ duration$_{Onset}$ and duration$_{Recovery}$ in the Huaibei Plain of 14 ~ 20 days and 8 ~ 20





days, respectively. The findings of the previous studies varied slightly from the FD durations in this study.

In this study, the duration$_{Total}$ of FD$_{RZSM}$, FD$_{SESR}$, and FD$_{MESR}$ were all found to be higher in the southern and eastern NCP and lower in the northern. The spatial distributions of the FD$_{RZSM}$ and FD$_{SESR}$ severity were comparable to those of the FD$_{RZSM}$ and FD$_{SESR}$ duration$_{Total}$, with the spatial pattern of high in the southern and low in the northern NCP. Zhang et al. (2021) investigated the FD$_{RZSM}$ characteristics in the Gan River Basin from 1961 to 2018 and discovered similar spatial distribution

patterns between severity and duration as well.

The annual affected areas of FD$_{RZSM}$ and FD$_{MESR}$ in this study exhibit a significant decrease trend, but that of FD$_{SESR}$ exhibits a slowly increase. Overall, the FD ratio, duration, severity and intensity grades in the NCP show a decrease in both temporal and spatial variations, and the spatial distribution patterns of the duration$_{Total}$ and severity trends are comparable. Additionally, Chen et al. (2019) also showed a decreasing FD affected area in the United States between 2000 and 2017. Noguera et al.

(2021) revealed that the severity and frequency of FD decreased slowly in Spain from 1960 to 2020 as well. Zhang et al. (2021) discovered that the FD$_{RZSM}$ frequency and duration in the Gan River Basin exhibited a decreasing tendency from 1961 to 2018. Liu et al. (2020) and Deng et al. (2022) also observed that there was a notable decline in the FD$_{RZSM}$ and FD$_{SESR}$ affected areas in the NCP region from 1981 to 1999 and 2000 and 2017, respectively. Furthermore, Christian et al. (2023) revealed a noteworthy reduction in FD$_{SESR}$ frequency in NCP from 1950 to 2015 as well. Studies have also indicated that there are other

FD characteristics trends that cannot be disregarded (Yuan et al., 2019, 2023; Zhang et al., 2022a; Zhang et al., 2022c), which are concerning with the FD identification methods, study areas, data sources, and study periods (Liu et al., 2020).

**4.2 Attribution analysis of the frequency difference between FD$_{SESR}$ and FD$_{MESR}$**

Compared to FD$_{SESR}$, the FD$_{MESR}$ frequency shows high spatial heterogeneity, which is connected to the frequency distribution of SESR$_{50th}$ and ΔSESR$_{50th}$. Fig.S7 illustrates the spatial distribution of average SESR$_{50th}$ and ΔSESR$_{50th}$. The SESR$_{50th}$ and

ΔSESR$_{50th}$ in NCP is all larger than 0, which makes it possible that SESR$_{40th}$ cannot indicate the low evaporative stress, as well as the underestimation of evaporative stress and inaccurate capture of FD events that would not occur. Therefore, it may result in the overestimation in the NCP (see Sect.3.2). In FD$_{SESR}$ identification, the interplay between the overestimation from SESR$_{50th}$ and the overestimation of ΔSESR$_{50th}$ in the NCP may result in a more pronounced overestimation of the FD$_{SESR}$ frequency due to the joint effect of SESR$_{50th}$ and ΔSESR$_{50th}$. As a result, the frequency of FD$_{MESR}$ is less than that of FD$_{SESR}$ in

the NCP.

The FD identification based on SESR and MESR differs in that MESR is used instead of SESR and is fitted with various PDFs rather than EDF, and the variable thresholds are utilized in the FD$_{MESR}$ identification. MESR and SESR are both linearly converted from ESR in order to facilitate comparisons of the FD identification results between different regions. Therefore, the difference between FD$_{SESR}$ and FD$_{MESR}$ can be traced back to two aspects: PDFs fitting and variable thresholds. To

demonstrate their contribution to the difference between FD$_{SESR}$ and FD$_{MESR}$, the thresholds in FD$_{SESR}$ method were referred to. The fixed thresholds, MESRonset1 = 40, MESRonset2 = 20, ΔMESRonset = 40, and MaxMESRchange = 25, were applied in the FD$_{MESR}$ method, which is called FD$_{MESR-invariable}$. Fig.S8 (a) ~ (c) displays the frequency of FD$_{MESR-invariable}$ and its difference from that of FD$_{SESR}$ and FD$_{MESR}$. Fig.S8 (d) (e) also show the contribution of PDFs fitting and variable thresholds to the differences between FD$_{SESR}$ and FD$_{MESR}$, respectively, as well as the relative contributions in Fig.S8 (e). FD$_{MESR-invariable}$

and FD$_{MESR}$ have a comparable frequency spatial distribution, with higher frequency in the north and lower in the south of NCP. Besides, FD$_{MESR-invariable}$ has a lower frequency than FD$_{SESR}$ in NCP, whereas a higher frequency than FD$_{MESR}$ in the southern NCP and a lower frequency than FD$_{MESR}$ in the northern NCP. According to Fig.S8 (d) (e), except for the small region in the northeastern NCP, the PDFs fitting contributes negatively to the difference between FD$_{SESR}$ and FD$_{MESR}$, while the variable thresholds contribute positively. Compared with the variable threshold contributions, the relative contribution of the

PDFs fitting to the variable thresholds mostly ranges from -1 to 0 in the northern NCP and from 0 to 0.5 in the southern NCP. Considering the negative contribution of PDFs fitting in the NCP, and the mostly absolute values of the relative contributions



are less than 1, it can be believed that the contribution of PDFs fitting is greater than that of variable thresholds. Therefore, the frequency difference between $FD_{SESR}$ and $FD_{MESR}$ is mostly due to the PDFs fitting.

### 4.3 Hotspot identification

Christian et al. (2023) classified regions with the FD frequency higher than 30% as hotspots. Nevertheless, this identification is mostly biased. Sreeparvathy and Srinivas (2022) comprehensively examined the FD frequency, duration, severity, and exposure risk characteristics, constructed a meteorological flash drought index (MFDI), and identified hotspots regions with MFDI higher than 100. The hotspot identification indicator in this study is derived by taking the average of the frequency, severity, and reciprocal of the intensity from the three FD identification methods. It may be said that FD events occur frequently,

seriously and intensely in regions with high hotspot indicators. The FD hotspots in this study are the regions where the hotspot indicator exceeds the 60th percentile. Through the comparison with typical historical events in Sect.3.4, the three FD identification methods have shown well performance in different typical events. Therefore, the hotspot identification indicator constructed in this study is based on these three methods simultaneously. This hotspot identification indicator considers the frequency, duration, and severity of FD, and comprehensively encompasses FD caused by insufficient precipitation and

excessive evaporation demand. Furthermore, the regional FD hotspot indicators was quantified using the percentiles of FD hotspot indicators, which can identify hotspots effectively. It is possible to export this FD hotspots identification threshold to other regions as well.

Meanwhile, this study only demonstrated the performance of the three FD methods through three typical historical events in Sect.3.4, and explored the rationality of the differences between $FD_{SESR}$ and $FD_{MESR}$ through the theoretical analysis. However,

this study did not investigate the applicable regions and conditions of the three methods, which is also one of the reasons why all the three methods are considered simultaneously in the hotspot identification. This defect requires further in-depth research in the future.

### 4.4 Thresholds' effects on the FD frequency

The effects of all thresholds of the three FD identification methods on the FD event identification are examined in this study.

While the spatial patterns of the FD frequency under various thresholds are similar, there are notable differences. The effects of thresholds on FD frequency are in accordance with their role in FD identification. The standard thresholds for $FD_{MESR}$ identification are set based on the $FD_{SESR}$ identification. Because of the sensitivity of thresholds to $FD_{MESR}$ frequency, finding the appropriate thresholds is essential to identify $FD_{MESR}$. Therefore, more studies are necessary to determine the appropriateness of the thresholds by an objective assessment.

### 540  5 Conclusions

Based on RZSM, SESR and MESR, the FD events in the NCP from 1981 to 2022 were identified, the FD characteristics, such as frequency, duration, severity and intensity, were revealed, and the temporal and spatial trend of FD characteristics were investigated. The FD hotspots in the NCP were illustrated likewise, as well as how the FD identification thresholds affect the FD identification. The main conclusions are as follows:

(1) The frequency distributions of $FD_{RZSM}$, $FD_{SESR}$ and $FD_{MESR}$ are high in the northern NCP and low in the southern, but the $duration_{Total}$ is high in the southern and eastern NCP and low in the northern. Similarly, the southern NCP experienced higher $duration_{Onset}$ and the northern NCP experienced lower, as well as the $duration_{Recovery}$ for $FD_{RZSM}$. There are notable differences in the severity spatial distribution of the three FD methods. The $FD_{RZSM}$ severity is high in the southwestern and eastern NCP, but low in the central and northern. $FD_{SESR}$ has a high severity in the southern NCP and a low severity in the northern NCP. In

contrast, $FD_{MESR}$ has a high severity in the northern and central NCP and low in the southern. The central NCP has larger



$FD_{RZSM}$ intensity values and lower $FD_{SESR}$ intensity values. Whereas $FD_{MESR}$ has larger intensity in the north of the NCP and lower in the south. The higher the FD intensity, the longer the onset stage, the shorter the recovery stage, the shorter the duration$_{Total}$, and the smaller the severity. The areas affected by FD exhibit clear and similar seasonal characteristics, especially $FD_{RZSM}$. The $FD_{RZSM}$ and $FD_{SESR}$ grids under different intensity grades vary greatly with the season, while the seasonal

fluctuations of $FD_{MESR}$ are small. But the commonality is that NCP is prone to extreme and exceptional FDs.

(2) The annual variations and spatial distributions of FD affected area, duration, severity and intensity in the NCP show a declining tendency, and the spatial distribution of duration and severity trends are similar. Three hotspots with frequent and severe FD events in the NCP are identified based on FD event characteristics, which are located in the northwestern, eastern and southwestern NCP, respectively. They all have developed agricultural planting, high population and diversified human

activities.

(3) The impact of various FD thresholds on FD frequency varies. RZSM from deeper soil layers identified a lower $FD_{RZSM}$ frequency. 'RZSMpentad1' is not sensitive to $FD_{RZSM}$ identification, while smaller 'RZSMonset1', 'RZSMpentad2' and 'MinDuration', as well as greater 'RZSMonset2' and 'RZSMtermination', are connected with a greater $FD_{RZSM}$ frequency. Both 'SESRonset1' and 'ΔSESRonset' are insensitive for $FD_{SESR}$ frequency. Nevertheless, $FD_{SESR}$ frequency is higher for

smaller 'MinDuration' and 'SESRpentad', and for larger 'SESRonset2' and 'MaxSESRchange'. The effects of 'MinDuration', 'MESRonset2', 'MESRpentad', and 'MaxMESRchange' on $FD_{MESR}$ frequency are similar to those of $FD_{SESR}$, while lower 'MESRonset1' and greater 'ΔMESRonset' are connected with higher frequency.

**CRediT authorship contribution statement**

**Siyao Zhang:** Conceptualization, Methodology, Software, Formal analysis, Investigation, Writing - Original Draft,
Visualization. **Jianzhu Li:** Conceptualization, Data Curation, Writing - Review & Editing, Supervision. **Ting Zhang:** Conceptualization, Resources, Writing - Review & Editing, Project administration. **Ping Feng:** Writing - Review & Editing, Supervision, Funding acquisition.

**Declaration of Competing Interest**

The authors declare that they have no conflict of interest.

**Data availability**

Data will be made available on request.

**Acknowledgements**

This work was supported by National Natural Science Foundation of China (No. 52079086).

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
