# Peer review of "Flash drought characteristics based on three identification methods in the North China Plain, China."

_Hydrology and Earth System Sciences, 2024_

## Author Comment (AC1)

Dear editor:

On behalf of all the contributing authors, I would like to express our sincere appreciations of your letter and the constructive comments from Referee #1 concerning our article entitled "Flash drought characteristics based on three identification methods in the North China Plain, China". All the comments are very helpful for revising our paper. We have studied and discussed all the comments point-by-point carefully, and accordingly made substantial revisions to our paper. All the changes we have made were in the red-colored text. If the response to the latter comments has already been mentioned in the previous response, it is provided in the pink text without the detailed explanation. Our point-by-point responses to all the comments are provided below in the blue-colored texts.

\*\*\*\*\*\*\*\*\*\*\*\*\*\*\*\*\*\*\*\*\*\*\*\*\*\*\*\*\*\*\*\*\*\*\*\*\*\*\*\*\*\*\*\*\*\*\*\*\*\*\*\*\*\*\*\*\*\*\*\*\*\*\*\*\*\*\*\*\*\*\*\*\*\*\*\*\*\*

############ Major Comments

1. Uncertainty in Data Sources:

The study implemented reanalysis ET, PET, and SM data. Due to the uncertainty inherent in these datasets, it is challenging to confirm whether the proposed MESR methodology accurately captures flash drought events. While the authors evaluated the methodology using three historical drought events (1981, 1983, and 1989), the nature of these events (flash or conventional drought) remains unclear, and their characteristics are not provided. Comparing MESR performance with recent, well-documented flash drought events would strengthen the reliability of the findings.

**Response:** Thank you for the comments. The two typical FD events occurring in 2017 and 2019 are utilized to evaluate the applicability of three FD identification methods in Section 3.4 (see Section 3.4 in lines 481-502). The FD events in 2017 and 2019 are identified by soil moistures, and both are well recorded with the detailed development and evolution records in Xue (2023) and Yao et al. (2022), further confirmed in Chen et al. (2024) as well. Grids suffering $FD_{RZSM}$, $FD_{SESR}$, and $FD_{MESR}$ are identified in the revised manuscript and compared with the development and evolution records of the two real FD events, demonstrating good consistency.

*"3.4 Typical historical events*

*To demonstrate the applicability of three FD identification methods, an evaluation is conducted based on two typical drought events occurring in 2017 and 2019 (Chen et al., 2024; Xue, 2023; Yao et al., 2022). Xue (2023) identified the FD events in the NCP between 1978 and 2020 using the soil moisture. It was found that the $FD_{RZSM}$ event began in late July 2017, terminated in the mid-August, and became serious in early August. Figure 9 (a) shows the spatial evolution of $FD_{RZSM}$, $FD_{SESR}$, and $FD_{MESR}$ from July to August 2017. $FD_{SESR}$ and $FD_{MESR}$ started in the southwestern NCP on July 5th, and alleviated on August 4th. After then, there were only sporadic FDs. A $FD_{RZSM}$ event occurred on July 15th and eased until August 9. After that, the affected area rapidly shrank and ended on August 29th. In late July, the affected area of $FD_{RZSM}$, $FD_{SESR}$, and $FD_{MESR}$ were all large, indicating a serious FD. Therefore, it can be considered that the $FD_{RZSM}$, $FD_{SESR}$, and $FD_{MESR}$ in 2017 in this study agree with the findings of Xue (2023).*

*Furthermore, the FDSESR and FDMESR started and developed before FDRZSM, indicating that they somewhat spread towards FDRZSM.*

[Figure]

[Figure]

**Figure 9 The spatiotemporal evolution process of FD events in (a) 2017 and (b) 2019.**

*Yao et al. (2022) discovered that FDRZSM in 2019 rapidly developed from April 30th to June 9th, during which the RZSM percentiles decreased sharply from 86% to 25%. Afterwards, the RZSM percentile decreased once again, and the FDRZSM severity peaked in July and recovered in August. However, Fig.9 (b) shows that the FDRZSM started on April 26th and recovered for a short time on June 5th, but it worsened again since June*

*15th with the largest affected area from late June to early July, then gradually recovered and terminated in August. $FD_{SESR}$ and $FD_{MESR}$ exhibited a similar evolution as $FD_{RZSM}$. It began on April 26th, recovered from May 31st, then continued to develop on June 15th, eased on July 10th, and ended on July 30th. The evolution is comparable to that from Yao et al. (2022). Therefore, the FD identification by RZSM, SESR, and MESR in this study might be in good agreement with the actual FD events, and the findings are trustworthy."*

2. Spatial Heterogeneity and Climate Regimes:

The study area is semi-humid, and identifying flash droughts requires consideration of background aridity and land cover impacts. One concern is the spatial heterogeneity in FD frequency detected by MESR, with significant differences between adjacent pixels scattered across the area, and such patterns are not evident in the other two methods. Evaluating MESR's performance in different climate regimes, such as semi-arid or sub-humid regions, using the Aridity Index (AI), would improve the robustness and generalizability of this research.

**Response:** Thank you for the comments. The spatial heterogeneity of the FD characteristics in the original manuscript has been analyzed, combined with the land use types, Aridity Index (AI), and the ratio of mean annual ET and PET in the revised manuscript (see lines 321-334, lines 361-364, lines 381-386, lines 398-405).

Furthermore, the performance of MESR has been evaluated in Section 4.3 in the revised manuscript (see lines 616-647). The coefficients of determination ($R^2$) are used to measure the capacity to explain the variance in the dependent variable by the linear regression between the independent and dependent variables. Considering the certain relationship between $FD_{RZSM}$, $FD_{SESR}$, and $FD_{MESR}$, the relationships between $FD_{RZSM}$, $FD_{SESR}$, and $FD_{MESR}$ have been explored by the $R^2$. In Section 4.3, "RZSM ~ SESR", which represents the relationship between $FD_{RZSM}$ and $FD_{SESR}$, is determined by the $R^2$ via the linear regression between the RZSM percentile (dependent variable) and SESR percentile (independent variable) of $FD_{RZSM}$ pentads. Meanwhile, the "RZSM ~ MESR", "SESR ~ RZSM", "MESR ~ RZSM", "SESR ~ MESR", and "MESR ~ SESR" are determined as well. Besides, the impact of AI on the spatial distributions of $R^2$ has also been explored.

*"The frequency of $FD_{RZSM}$ is high in the central and northeastern NCP and low in the southern NCP with two high-frequency regions in the central and northeastern NCP. The AI values of central and northeastern NCP are 0.2 ~ 0.3 and less than 0.2, respectively (Fig.S1 (b)). Therefore, they are comparatively dry. The lack of precipitation and an increase in evapotranspiration would fasten the decline in soil moisture and increase the probability of FD in the central NCP than the southern NCP (Gou et al., 2022; Yuan et al., 2023). Besides, the northeastern NCP is woodland with high ET (Guo et al., 2007). Adding the high-water demand, it occurs frequent $FD_{RZSM}$. The frequency of $FD_{SESR}$ and $FD_{MESR}$ are both high in the north-central NCP and low in the northeastern and southern NCP and are opposite to the ratio of annual ET and PET (Fig.S1 (c)), indicating that a region with greater evaporative stress would encounter more $FD_{SESR}$ and $FD_{MESR}$. North-central NCP is cultivated land. The*

*evaporative stress of north-central NCP fluctuates significantly due to the influence of irrigation, which causes frequent $FD_{SESR}$ and $FD_{MESR}$. In contrast, the woodland in the northeastern NCP is usually not irrigated artificially, and the evaporative stress is mostly influenced by climate conditions. Therefore, northeastern NCP has less evaporative stress fluctuations and low $FD_{SESR}$ and $FD_{MESR}$ frequency (Guo et al., 2007). Southern NCP has higher temperature, greater evapotranspiration, and more abundant precipitation as latitude decreases. The balanced hydrothermal conditions lead to a greater AI and lower FD frequency in the southern NCP."*

[Figure]

***Figure S1 Spatial distribution of (a) the land use in 2010, (b) AI, and (c) the ratio of annual ET and PET in NCP.***

*"Warmer temperature may result in longer FD duration (Zhang et al., 2022c). Woodlands take longer to recover from drought than the cultivated lands (Wu et al., 2024). Additionally, human activities might also have an impact on the FD duration$_{Total}$. Irrigation might significantly alleviate FD in cultivated land, but the woodland in the northeastern NCP is less impacted by human activity and might have a longer FD duration$_{Total}$."*

*"Warming not only lengthens the drought durations of the southern NCP, but also exacerbates them by increasing surface evapotranspiration losses and decreasing the soil moistures (Yuan et al., 2019; Zhang et al., 2021). The long duration might also result in the great severity. Whereas $FD_{MESR}$ has high severity in the northern and central NCP and low severity in the southern NCP. The spatial distribution of the $FD_{MESR}$ severity is opposite to the ratio of annual ET and PET, which is lower in the northern NCP but higher in the southern (Fig.S1 (c)). It illustrated that the duration$_{Total}$ and severity of $FD_{MESR}$ have a stronger correlation with the evaporative stress than that of $FD_{SESR}$."*

*"For $FD_{RZSM}$, RZSM percentiles decrease slower in west-central NCP than in other regions. Even though the west-central NCP is in the arid state with a low AI of 0.2 ~ 0.4, it might be because the west-central NCP is cultivated land and irrigation has a significant impact on the soil moisture, which might effectively alleviate the decline in RZSM. Although the AI in northern NCP is less than 0.4 as well, the woodland in this region is less impacted by human activities like irrigation, which causes RZSM to rapidly decrease. Additionally, the high temperature and great ET in the southern NCP hasten the RZSM decline rate. For $FD_{SESR}$ and $FD_{MESR}$, even though southern NCP has great ET, the PET is great as well, which might not lead to a low ESR and high evaporative stress (Fig.S1). Abundant precipitation and low evaporative stress ease the declining rate of SESR and MESR."*

*"**4.3 Explanatory ability between different FD types***

*Given the impact of climate control on the FD occurrence (Mukherjee and Mishra, 2022), there might be a certain relationship between different FD types. Therefore, the relationships among $FD_{RZSM}$, $FD_{SESR}$, and $FD_{MESR}$ are explored by the coefficient of determination ($R^2$) which stands for the capacity to explain the variance in the dependent variable by the linear regression between the independent and dependent variables (Mukherjee and Mishra, 2022). In particular, the relationship between $FD_{RZSM}$ and $FD_{SESR}$, which is referred to as "RZSM ~ SESR", is represented by the $R^2$ determined via the linear regression between the RZSM percentile (dependent variable) and SESR percentile (independent variable) of $FD_{RZSM}$ pentads. Meanwhile, the "RZSM ~ MESR", "SESR ~ RZSM", "MESR ~ RZSM", "SESR ~ MESR", and "MESR ~ SESR" are determined, as shown in the first two columns in Fig.S15.*

*In Fig.S15 (a) and (b), both "RZSM ~ SESR" and "RZSM ~ MESR" explain more than 40% of the variance in RZSM percentile in the central NCP, but less than 30% in the other regions. In Fig.S15 (e) and (f), the southern NCP has more "SESR ~ RZSM" explaining the variance in SESR and more "MESR ~ RZSM" explaining the variance in MESR percentile (mostly about 15% ~ 25%) than the northern (less than 15%). Whereas "SESR ~ MESR" explains over 90% of the variance in SESR, as well as "MESR ~ SESR" explains the variance in MESR, as shown in Fig.S15 (i) and (j). Overall, the explanatory ability from high to low is: "SESR ~ MESR" and "MESR ~ SESR" > "RZSM ~ SESR" and "RZSM ~ MESR" > "SESR ~ RZSM" and "MESR ~ RZSM". SESR and MESR are all based on the linear transformation of ESR, which makes them good in explaining each other with a high $R^2$ of over 90%. The relationship between MESR and RZSM ("RZSM ~ MESR" and "MESR ~ RZSM") is quite comparable to that between SESR and RZSM ("RZSM ~ SESR" and "SESR ~ RZSM"), highlighting the reliability of FD identification based on MESR. The differences in Fig.S15 (c), (g), and (k) further demonstrate the similarities between SESR and MESR as well.*

*The spatial distributions of $R^2$ also point to the sensitivity to the AI, as shown in Fig.S15 (d) (h) and (l). For "RZSM ~ SESR" and "RZSM ~ MESR", the variance explanatory ability increases with the increasing AI in the region with AI below 0.3, but decreases in the region with AI above 0.3. In the region where AI is between 0.2 and 0.3, they explain the greatest variance in the RZSM percentile (about 60%). Overall, the RZSM percentiles could be explained more by the SESR and MESR percentiles in the dryer region with the exception of the region with AI less than 0.2, which might be related to that RZSM is greater initially in the wet region with extended memory (Mukherjee and Mishra, 2022). For "SESR ~ RZSM" and "MESR ~ RZSM", the variance explanatory ability is less than 20%, and it is obviously greater in the region with AI more than 0.2 than in the region with AI less than 0.2. SESR and MESR percentiles could be better explained by the RZSM percentile in the wetter region with less evaporative stress and higher evaporation. The $FD_{SESR}$ and $FD_{MESR}$ belonging to meteorological drought might lead to the $FD_{RZSM}$ belonging to agricultural drought, making that $R^2$ of "SESR ~ RZSM" and "MESR ~ RZSM" is lower than that of "RZSM ~ SESR" and "RZSM ~ MESR". For "SESR ~ MESR" and "MESR ~ SESR", the explanatory ability exceeding 90% increases with the increasing AI overall."*

[Figure]

*Figure S15 Spatial distribution of the $R^2$ determined by (a) "RZSM ~ SESR", (b) "RZSM ~ MESR", (e) "SESR ~ RZSM", (f) "MESR ~ RZSM", (i) "SESR ~ MESR", and (j) "MESR ~ SESR", as well as the differences of $R^2$ (c) between "RZSM ~ SESR" and "RZSM ~ MESR", (g) between "SESR ~ RZSM" and "MESR ~ RZSM", and (k) between "SESR ~ MESR" and "MESR ~ SESR". The boxplots in the (d), (h), and (l) illustrate the $R^2$ in the "RZSM ~ SESR" and "RZSM ~ MESR", "SESR ~ RZSM" and "MESR ~ RZSM", and "SESR ~ MESR" and "MESR ~ SESR" over different AI values.*

3. Justification of Methodology:

The paper lacks a clear explanation for multiplying ESR values by their mean (climatological or long-term) to create MESR. This difference appears to be a primary factor distinguishing MESR from SESR in terms of frequency. The authors should further clarify and justify this decision in the main text for better understanding and transparency.

**Response:** Thank you for the comments. $FD_{MESR}$ is proposed based on the limitations of $FD_{SESR}$ in lines 84-92. The first one is that whether ESR follows a normal distribution requires further determination, which makes the rationality of normalizing ESR into SESR is still up for debate. The second one is that the $SESR_{40th}$ and $\Delta SESR_{40th}$ might be greater than 0. Then we have added the detailed reasons for using MESR instead of SESR to identify FD in lines 198-209. There are three main differences between $FD_{MESR}$ and $FD_{SESR}$ identification. The first one is that the ESR value is divided by its mean to construct the MESR series instead of normalizing the ESR to create SESR. The second one is that the percentiles of MESR and ΔMESR for each pentad are fitted using the optional PDF instead of utilizing EDF to convert into percentiles. And the third one is the variable thresholds of MESR and ΔMESR are employed in the process of FD identification.

*"However, the SESR application has some problems. When identifying FD using SESR, both SESR and the change of SESR (ΔSESR) theoretically follow normal distributions, therefore the 50th percentile ($SESR_{50th}$ and $\Delta SESR_{50th}$) equal 0, indicating that*

*$\Delta SESR_{50th}$ denotes no change in SESR ($\Delta SESR = 0$), whereas $\Delta SESR$ below the 40th percentile of $\Delta SESR$ values ($\Delta SESR_{40th}$) denotes a declining in SESR. However, whether SESR and $\Delta SESR$ follow a normal distribution remains to be determined, and if not, the $\Delta SESR_{40th}$ may not be less than 0. In the study by Gou et al. (2022), it was found that the 36th percentile of $\Delta SESR$ ($\Delta SESR_{36th}$) corresponds to an increase in SESR, where $\Delta SESR_{36th}$ is greater than 0. This phenomenon may be because that SESR gradually decreases during periods without precipitation, and increases during the precipitation process. It is conceivable for $\Delta SESR$ to be less than 0 during periods with no precipitation and larger than 0 during precipitation periods. Consequently, the FD events may be underestimated or overestimated."*

*"There are three main differences between MESR and SESR identification. Firstly, the ESR value is divided by its mean to construct the MESR series instead of normalizing the ESR to create SESR. Regardless of whether ESR is standardized as SESR or MESR, their percentile values are unaffected by the linear transformations based on ESR. However, whether ESR follows a normal distribution requires further determination, which makes the rationality of normalizing ESR into SESR is still up for debate. Whereas, when dividing ESR by its mean and transforming it into MESR, it is not necessary to take into account the PDF that ESR follows. Secondly, the percentiles of MESR and $\Delta MESR$ for each pentad are fitted using the optional PDF instead of utilizing EDF to convert into percentiles. Since the distribution function that MESR and $\Delta MESR$ follow is yet unknown, several PDFs are fitted in order to select the best PDF, which can produce more precise percentiles. Lastly, the variable thresholds of MESR and $\Delta MESR$ are employed in the process of FD identification. Due to the uncertainty surrounding whether SESR and $\Delta SESR$ follow a normal distribution, as well as the phenomenon that the $SESR_{40th}$ and $\Delta SESR_{40th}$ might be greater than 0 (as found in Gou et al. (2022)), the variable percentiles of MESR and $\Delta MESR$ are used as the threshold for FD identification to make sure the threshold is less than 0."*

#############Minor Comments

4. Lines 26-27: Replace "becomes" with "become" to align with the plural subject "droughts."

**Response:** Thank you for the comments. We have revised "becomes" into "become" (see lines 26-27).

*"The terrestrial water cycle accelerates and droughts become more frequent under global warming."*

5. Line 40: Remove "respectively" for clarity.

**Response:** Thank you for the comments. The "respectively" has been deleted in the revised manuscript (see lines 38-40).

*"Currently, the FD identification methods can be mainly divided into three categories based on conventional drought indicators, soil moisture, and atmospheric evaporation demand."*

6. Lines 84-85: SESR is a method for identifying flash droughts, so it is better to use

the word 'using' in this sentence. Here is the revised version:

"However, the SESR application has some problems. When identifying FD using SESR, both SESR and the change in SESR"

**Response:** Thank you for the comments. We have revised the presentation (see lines 84-87).

*"However, the SESR application has some problems. When identifying FD using SESR, both SESR and the change of SESR ($\Delta SESR$) theoretically follow normal distributions, therefore the 50th percentile ($SESR_{50th}$ and $\Delta SESR_{50th}$) equal 0, indicating that $\Delta SESR_{50th}$ denotes no change in SESR ($\Delta SESR = 0$), whereas $\Delta SESR$ below the 40th percentile of $\Delta SESR$ values ($\Delta SESR_{40th}$) denotes a declining in SESR."*

7. Line 96: It would be better to rephrase this sentence for better clarification, and start with 'In this study, ... ' Here is the revised version:

In this study, a new method based on the ......

**Response:** Thank you for the comments. We have revised the presentation (see lines 96-97).

*"In this study, a new method based on the multiples of the mean evaporative stress ratio (MESR) for FD identification has been developed to address the aforementioned problems in SESR identification method."*

8. Line 111: One of the most important factors in characterizing droughts is considering background aridity. One of my concerns regarding this paper is that the study area is mainly a semi-humid region. The baselines of SM percentile or SESR vary across different climate regimes.

**Response:** Thank you for the comments. The thresholds in the FD identification on one grid are the specific percentiles (like $40^{th}$ or $20^{th}$ percentiles) of RZSM, SESR, or MESR on the specific grid for the specific pentad throughout the entire study period. Although the percentiles of the FD thresholds are fixed for all grids ($RZSM_{40th}$, $RZSM_{20th}$, $SESR_{40th}$, $SESR_{20th}$, $\Delta SESR_{40th}$, $MESR_{Pr1-10th}$, $MESR_{Pr1-30th}$, and $\Delta MESR_{Pr2-10th}$), the determination of the percentiles is based on the RZSM, SESR, and MESR series on the specific grid for the specific pentad during the whole study period. It can be considered that the FD thresholds are temporally and spatially influenced. Besides, SESR and MESR are both obtained through the linear transformation of ESR, which does not change their percentiles. The $40^{th}$ and $20^{th}$ percentiles of RZSM and SESR, as well as the $(Pr1-10)^{th}$ and $(Pr1-30)^{th}$ percentiles of MESR, indicate a low level of soil moisture and evaporative stress value. Meanwhile, the $\Delta SESR_{40th}$ and the $\Delta MESR_{(Pr2-30)th}$ indicate a significant decreasing in the SESR or MESR. Therefore, the thresholds for FD identification in different climate regions do not affect the FD identification results, and these thresholds could be widely applied in various climate regions.

9. Lines 120-127: Soil moisture, ET, and PET datasets used in this study are reanalysis and there is uncertainty in these data sets. Additionally, different reanalysis datasets have great differences in their values, so this study can benefit from using different SM, ET, and PET datasets. GLEAM, or MERRA2?

**Response:** Thank you for the comments. The soil moisture, ET, and PET obtained from the GLEAM and GLDAS 2 datasets are detailed in lines 133-145. Based on the GLEAM and GLDAS 2 datasets, the uncertainties of the different reanalysis datasets have been evaluated in Section 4.1 of the revised manuscript (see lines 552-574). We have analyzed the bias of the pentad RZSM, SESR, and MESR percentiles from 1981 to 2022 between ERA5-Land and GLEAM and GLDAS 2 datasets by Taylor diagrams in Figure S11. The spatial distribution of the pentad RZSM, SESR, and MESR percentile correlations between ERA5-Land and GLEAM and GLDAS 2 datasets in Figure S12 have also shown great similarities. The FD characteristics identified based on ERA5-Land, GLEAM, and GLDAS 2 datasets have also been compared in Figures S5, S13 and S14. The similarity of pentad RZSM, SESR, and MESR percentiles from various datasets, as well as the FD characteristics based on various datasets, effectively demonstrates the reliability of findings.

*"Two additional reanalysis datasets, fourth version of the Global Land and Evaporation Amsterdam Model (GLEAM v4.1a; Miralles et al., 2011) and Global Land Data Assimilation System version 2 (GLDAS 2; Beaudoing &Rui, 2019; Beaudoing &Rui, 2020) were introduced to evaluate the uncertainty of FD identified by the ERA5-Land dataset. The surface (0 ~ 10 cm) soil moisture (unit: $m^3\ m^{-3}$), root-zone (0 ~ 100 cm) soil moisture (RZSM, unit: $m^3\ m^{-3}$), actual evaporation (E, unit: $mm\ day^{-1}$), and potential evaporation (PET, unit: $mm\ day^{-1}$) provided by GLEAM v4.1a dataset are on a 0.1°×0.1° latitude-longitude grid and with a daily temporal resolution (Xue and Wu, 2024). Evapotranspiration (ET, unit: $kg\ m^{-2}\ s^{-1}$), potential evaporation rate (unit: $W\ m^{-2}$), and soil moistures from 0 ~ 10 cm, 10 ~ 40 cm, and 40 ~ 100 cm (unit: $kg\ m^{-2}$) derived from the GLDAS 2 dataset are obtained at 0.25°×0.25° spatial resolution and 3-hourly temporal resolution. The potential evaporation rate was transformed into the potential evaporation (PET) by calculating the accumulation over time. Similar to the ERA5-Land dataset, the soil moistures of layers 0 ~ 40 cm and 0 ~ 100 cm in GLDAS 2 were determined by the weighted average. Because the GLDAS 2.0 dataset spans between 1948 and 2014, while the GLDAS 2.1 dataset is from 2000, the GLDAS 2.0 dataset from 1981 to 1999 and the GLDAS 2.1 dataset from 2000 to 2022 were utilized in accordance with the method of Wang and Yuan (2021). The findings in this study are based on the ERA5-Land dataset unless otherwise noted."*

[Figure]

***Figure S11** Taylor diagram for the pentad (a) RZSM, (b) SESR, and (c) MESR percentiles based on ERA5-Land, GLEAM, and GLDAS 2 datasets.*

[Figure]

**Figure S12 Spatial distribution of the Pearson correlation of the pentad RZSM, SESR, and MESR percentiles between ERA5-Land and (a) ~ (c) GLEAM and (d) ~ (f) GLDAS 2 datasets.**

*"4.1 Uncertainties from the reanalysis datasets*

*To explore the data-related uncertainties, the soil moisture, ET, and PET obtained from two additional reanalysis datasets, GLEAM and GLDAS 2 datasets, are utilized to identify the $FD_{RZSM}$, $FD_{SESR}$, and $FD_{MESR}$. Due to that the RZSM, SESR, and MESR are the basis for FD identification, the pentad percentiles of RZSM, SESR, and MESR are determined. Figure S11 illustrates the Taylor diagrams of the pentad RZSM, SESR, and MESR percentile series from 1981 to 2022 for ERA5-Land, GLEAM, and GLDAS 2 datasets, with the ERA5-Land dataset as the observation. The points representing the pentad SESR and MESR percentiles from GLEAM and GLDAS 2 datasets are all very close. The correlation coefficients between pentad RZSM percentile and SESR or MESR percentiles are around 0.7, the centered root mean square differences are around 0.8, and the standard deviations are approximately 1. Therefore, the pentad percentiles of RZSM, SESR, and MESR of the ERA5-Land dataset are similar to those of GLEAM and GLDAS 2. Figure S12 displays the spatial distribution of the pentad RZSM, SESR, and MESR percentile correlations between ERA5-Land and GLEAM and GLDAS 2 datasets. As shown in Fig.S12 (a) and (d), the RZSM percentile correlation between ERA5-Land and GLEAM is comparable to that between ERA5-Land and GLDAS 2, both of which are primarily greater than 0.6. While the SESR percentile correlation between ERA5-Land and GLEAM is similar to the MESR percentile correlation, which both exceeds 0.5 mostly. The correlation of SESR and MESR percentiles between ERA5-Land and GLDAS 2 is mostly between 0.4 and 0.7, with a comparable spatial distribution pattern. The higher correlation between ERA5-Land and GLEAM than between ERA5-Land and GLDAS 2 might be due to the coarse spatial resolution of GLDAS 2.*

[Figure]

**Figure S5 Histogram of FD characteristics identified by RZSM, SESR, and MESR based on ERA5-Land.**

[Figure]

**Figure S13 Same as Figure S5, but based on GLEAM.**

[Figure]

**Figure S14 Same as Figure S5, but based on GLDAS 2.**
Besides, the FD characteristics identified based on ERA5-Land, GLEAM, and GLDAS

*2 datasets are displayed in Figs.S5, S13 and S14. The distributions of the $FD_{RZSM}$,*
*$FD_{SESR}$, and $FD_{MESR}$ characteristics based on various datasets are comparable with the*
*exception of the $FD_{MESR}$ intensity based on GLDAS 2. Meanwhile, the proportions of*
*various FD grades determined by intensity from diverse data sources also demonstrate*
*an indisputable resemblance, as seen in Fig.S6. The similarity of pentad RZSM, SESR,*
*and MESR percentiles from various datasets, as well as the FD characteristics based*
*on various datasets, effectively demonstrates the reliability of our findings.*

10. Lines 182-183: what is the main reason for dividing ESR value by its mean?
Although there are some benefits to doing this, I am concerned that normalizing ESR
by the mean could reduce the sensitivity of the method in detecting flash droughts,
especially in periods or regions with naturally higher evapotranspiration stress. The
underlying issue with normalizing ESR by its mean is that it reduces the relative
magnitude of $MESR_0$ values when the baseline ESR is high, effectively making it
harder to detect rapid changes. If the authors have any explanation for this, it would be
helpful to include it in the main text.

**Response:** Thank you for the comments. In the process of identifying FD based on
SESR and MESR, the first step is the percentile transformation, followed by the FD
identification by the FD identification criteria that are related to the percentiles. The
thresholds of FD identification based on SESR and MESR are both related to the
percentiles of SESR and MESR. In the process of standardizing ESR into SESR or
MESR, a linear transformation is conducted on ESR, which would not influence the
corresponding percentiles. The percentiles are determined by the PDF that ESR fits.
ESR might not follow a normal distribution, making that the rationality of normalizing
ESR into SESR needs further discussion. Whereas, standardizing ESR by its mean and
converting it into MESR does not involve the problem of whether ESR follows a normal
distribution. Therefore, ESR is divided by its mean and standardized as MESR. The
reason for using MESR instead of SESR to identify FD has been added in lines 198-
209, which has been mentioned in Comment 3. Therefore, it can be considered that the
linear transformation methods conducted on ESR would not affect the sensitivity of FD
identification, and using the mean ESR for standardization would not affect the FD
identification based on the percentiles. The detailed reason that why normalizing ESR
by the mean would not reduce the sensitivity of FD identification has also been
supplemented in lines 248-252.

*"In the process of $FD_{SESR}$ and $FD_{MESR}$ identification, the first step is converting the*
*SESR, $\Delta SESR$, $MESR_0$, and $\Delta MESR_1$ into percentile values, and then identify FD by the*
*criteria related to the percentiles. Since the linear transformation applied to the ESR*
*would not change the percentiles, the linear transformation methods would not have an*
*impact on the percentiles of SESR, $\Delta SESR$, $MESR_0$, and $\Delta MESR_1$ and the thresholds of*
*the FD criteria, as well as the FD identification results. It is said that the FD*
*identification based on the percentiles would be unaffected by using the mean ESR for*
*standardization."*

11. Line 184: It is climatological mean or long-term mean?

**Response:** Thank you for the comments. $\overline{ESR}$ is the climatological mean value of ESR at a specific grid for a specific pentad for all years during the study period (see lines 214-215).

"where $\overline{ESR}$ is the climatological mean value of ESR at a specific grid for a specific pentad for all years during the study period, and $MESR_0$ is the multiple of $\overline{ESR}$ at the specific grid for a specific pentad."

12. Line 184-191: Specify whether the mean is climatological or long-term.
**Response:** Thank you for the comments. The mean is climatological mean (see lines 222-224).

*"where $\Delta MESR_0$ is the change in MESR between adjacent pentads, $\overline{\Delta MESR_0}$ is the climatological mean value of $\Delta MESR_0$ at the specific grid for a specific pentad over the whole study period, and $\Delta MESR_1$ is the multiple of $\overline{\Delta MESR_0}$ for the pentad o all study period."*

13. Line 191: Again, climatological mean or long-term mean?
**Response:** Thank you for the comments. The mean is climatological mean (see lines 222-224 in Comment 12).

14. Line 205: Likely a typo; change FD (SESR) to FD (MESR).
**Response:** Thank you for the comments. It is a typo. We have revised "$FD_{SESR}$" into "$FD_{MESR}$" (see line 237).
"$FD_{MESR}$ can be identified by the following four criteria:"

15. Line 213: This sentence seems to be uncompleted!
**Response:** Thank you for the comments. We have corrected the original manuscript and removed this sentence (see line 245).

16. Line 242: Clarify that "mean value" refers to a single point, and specify whether it is climatological or long-term.
**Response:** Thank you for the comments. The $frequency_0$ is the FD frequency from 1981 to 2022 in the NCP on the grid; the $severity_0$ and $intensity_0$ are the average severity and intensity from 1981 to 2022 in the NCP on the grid. The "mean value" is long-term mean, and the detailed explanations for the "mean value" have been supplemented in lines 273-286.

$$frequency' = (frequency_0 - frequency_{min}) / (frequency_{max} - frequency_{min}) \tag{9}$$

$$severity' = (severity_0 - severity_{min})/(severity_{max} - severity_{min}) \tag{10}$$

$$intensity' = (intensity_0 - intensity_{min})/(intensity_{max} - intensity_{min}) \tag{11}$$

$$\overline{frequency'} = 0.5frequency'_{RZSM} + 0.25frequency'_{SESR} + 0.25frequency'_{MESR} \tag{12}$$

$$\overline{severity'} = 0.5severity'_{RZSM} + 0.25severity'_{SESR} + 0.25severity'_{MESR} \tag{13}$$

$$\overline{intensity'} = 0.5intensity'_{RZSM} + 0.25intensity'_{SESR} + 0.25intensity'_{MESR} \tag{14}$$

$$Hotspots = \left(\overline{frequency'} + \overline{severity'} + 1/\overline{intensity'}\right)/3 \tag{15}$$

*where the $frequency_0$ is the FD frequency from 1981 to 2022 in the NCP on the grid; the $severity_0$ and $intensity_0$ are the average severity and intensity from 1981 to 2022 in the NCP on the grid; the $frequency_{max}$, $frequency_{min}$, $severity_{max}$, $severity_{min}$, $intensity_{max}$,*

*and intensity$_{min}$ are the maximum and minimum values of the frequency$_0$, severity$_0$, and intensity$_0$ in the NCP; frequency', severity', and intensity' are the max-min normalization of the frequency$_0$, severity$_0$, and intensity$_0$; and the $\overline{frequency'}$, $\overline{severity'}$, and $\overline{intensity'}$ are the weighted average frequency', severity', and intensity' of FD$_{RZSM}$, FD$_{SESR}$, and FD$_{MESR}$. Due to that the FD$_{MESR}$ method is considered an extension of the FD$_{SESR}$ method, the weights for the characteristics of FD$_{RZSM}$, FD$_{SESR}$, and FD$_{MESR}$ are 0.5, 0.25, and 0.25, respectively."*

17. Lines 259-264: Although the discussion in this paragraph is statistically correct, I believe an important point has been overlooked: the most critical aspect of flash droughts is their rapid onset and intensification. As long as the SESR method can detect this characteristic of flash droughts, the exact value of the 50th percentile is less significant, particularly in humid or hyper-humid climate regimes where the ESR baseline might be higher than in arid or semi-arid regions. In regions with lower background aridity, the 50th percentile of SESR might be higher than zero, primarily due to dense vegetation. In such cases, it is crucial to capture the rapid reduction in SM/ESR that leads to flash drought.

Mukherjee and Mishra (2022) demonstrated that using different indicators, such as ESR and soil moisture, to identify flash droughts results in varying frequencies across different climate regimes. This is an important factor to consider, especially in this study.

Mukherjee, S., & Mishra, A. K. (2022). Global Flash Drought Analysis: Uncertainties From Indicators and Datasets. Earth's Future, 10(6), e2022EF002660. https://doi.org/10.1029/2022EF002660

**Response:** Thank you for the comments. The rapid reduction in RZSM, SESR, or MESR that leads to flash droughts are judged by the percentiles of ΔRZSM, ΔSESR, or ΔMESR. If SESR follows normal distribution, the 50$^{th}$ percentile being equal to 0 represents the median of the SESR series, and the 40$^{th}$ percentile being less than 0 represents a low value in the SESR series. If ΔSESR follows normal distribution, the 50$^{th}$ percentile being equal to 0 represents no change in SESR, and the 40$^{th}$ percentile being less than 0 represents the decreasing SESR. Therefore, the 40$^{th}$ percentile of SESR represents a low ESR, and 40$^{th}$ percentile of ΔSESR represents a decreasing ESR. Figure 3 (b) illustrates that amount of ΔSESR$_{50th}$ is larger than 0, which might lead to the ΔSESR$_{40th}$ greater than 0, reflecting an increasing SESR. It represents a rapid decline in SESR theoretically, but in reality, represents the increasing in SESR, which might lead to the misjudgments of FD events. So does the ΔSESR$_{50th}$ less than 0. Moreover, the SESR and ΔSESR percentiles are both calculated on the grid by pentads, meaning that the SESR$_{50th}$ and ΔSESR$_{50th}$ on different grids and pentads have different SESR$_{50th}$ and ΔSESR$_{50th}$ values, and the SESR$_{50th}$ and ΔSESR$_{50th}$ are temporally and spatially affected (see lines 289-300).

*"To graphically illustrate the limitations of the FD$_{SESR}$ method, the distributions of SESR$_{50th}$ and ΔSESR$_{50th}$ are shown in Fig.3 (a) (b). The SESR and ΔSESR percentiles are both calculated on the grid by pentads, meaning that the SESR$_{50th}$ and ΔSESR$_{50th}$ on various grids and pentads have various SESR$_{50th}$ and ΔSESR$_{50th}$ values, and SESR$_{50th}$ and ΔSESR$_{50th}$ are temporally and spatially affected. Theoretically, SESR and ΔSESR follow the standard normal distribution, where SESR$_{50th}$ = 0 and ΔSESR$_{50th}$ = 0,*

*indicating that SESR$_{40th}$ is below the average level and ΔSESR$_{40th}$ is less than 0 with the decreasing SESR. However, the skewed distributions of SESR$_{50th}$ and ΔSESR$_{50th}$ in NCP demonstrate that SESR$_{50th}$ and ΔSESR$_{50th}$ are generally not 0. When SESR$_{50th}$ > 0, it is possible that SESR$_{40th}$ > 0, indicating that ESR$_{40th}$ exceeds $\overline{ESR}$ and SESR$_{40th}$ cannot indicate the low evaporative stress value. When ΔSESR$_{50th}$ > 0, maybe the corresponding ΔSESR$_{40th}$ > 0, reflecting an increasing SESR, which could result in the underestimation of evaporative stress value and inaccurate capture of FD events that would not occur. When SESR$_{50th}$ < 0, ESR$_{40th}$ may be significantly lower than $\overline{ESR}$, which would indicate a lower evaporative stress value. When ΔSESR$_{50th}$ < 0, ΔSESR$_{40th}$ is also significantly less than 0, indicating a severe decreasing SESR, leading to the overestimation of evaporative stress value and disregard for the FD events that actually occurred."*

18. Figure 4, first row: As shown in this figure, the frequency of events detected by MESR is significantly lower than that detected by SESR. It would be helpful if the authors could show how many events detected by MESR are similar to those detected by SESR. Since both methods use ESR, it would be beneficial to compare these methods. One possible approach would be to compare binary time series of drought events detected by these methods and calculate their overlap.

**Response:** Thank you for the comments. The rationality of the spatial distributions of FD$_{RZSM}$, FD$_{SESR}$, and FD$_{MESR}$ frequency has been analyzed in conjunction with the land use types, Aridity Index (AI), and the ratio of mean annual ET and PET (see Comment 2 and lines 321-334). Besides, the overlapping events between FD$_{SESR}$ and FD$_{MESR}$ have been detected in Figure S4. The great overlapping event proportion and the significant correlation relationship between the pentad SESR and MESR percentiles demonstrate the reliability of the FD$_{MESR}$ identification (see lines 347-354).

[Figure]

***Figure S4 The number of (a) FD$_{SESR}$, (b) FD$_{MESR}$, (c) overlapping FD$_{SESR}$ and FD$_{MESR}$ events, the proportion of the overlapping events in (d) FD$_{SESR}$ and (e) FD$_{MESR}$ events, as well as (e) the correlation between pentad SESR and MESR percentiles.***

*"FD events that exhibit temporal overlap between FD$_{SESR}$ and FD$_{MESR}$ are regarded as*

*overlapping events, and Fig.S4 shows the number of overlapping events between the two. According to Fig.S4 (a) and (b), the spatial distributions of the number of $FD_{SESR}$ and $FD_{MESR}$ events are similar to their frequencies. Furthermore, overlapping events have a similar spatial distribution to $FD_{SESR}$ events in Fig.S4 (c), which might be because the $FD_{SESR}$ events are less than $FD_{MESR}$ events. As can be seen in Fig.S4 (d), the overlapping FD events make up more than 50% and even more than 60% of $FD_{SESR}$ events with the exception of the northwestern NCP. Figure S4 (f) also shows that the correlation between the pentad SESR and MESR percentiles is around 1, demonstrating a strong linear relationship between the two. The large overlapping proportion and linear relationship display the reliability of the $FD_{MESR}$ identification."*

19. Line 305: Is 5 pentads for the flash drought onset stage not too long? Flash droughts are characterized by their rapid onset, and 5 pentads is not particularly rapid. I would suggest setting a limitation on the duration of the flash drought onset stage in this research, similar to the RZSM method, which has an onset duration limitation. Additionally, are Figures 4 and S1 showing the average duration? If so, this implies that in some events, the onset stage of the flash drought is longer than 5 pentads!

**Response:** Thank you for the comments. Unlike the $FD_{RZSM}$ method, which manages the onset speed by limiting the onset duration in criteria a), criteria d) of $FD_{SESR}$ ensures that the drought has a rapid development rate and would not be affected by the temporary moderation of SESR (see lines 195-196).
The spatial distribution maps of FD characteristics in Figures 4 and S1 are the mean characteristics of all FD events that occur grid by grid, therefore the characteristics for the FD events might be larger or smaller than the mean. We have determined the characteristics of all FD events during 1981 ~ 2022, and displayed the histograms in Figure S5. Mostly duration$_{Onset}$ of the $FD_{RZSM}$, $FD_{SESR}$, and $FD_{MESR}$ range from 2 to 4 pentads (see lines 372-378).

*"Unlike the $FD_{RZSM}$ method, which manages the onset speed by limiting the onset duration in criteria a), criteria d) of $FD_{SESR}$ ensures that the drought has a rapid development rate and would not be affected by the temporary moderation of SESR."*
*"Since the spatial distribution maps of FD characteristics in Fig.4 are the mean characteristics of all FD events that occur grid by grid, the characteristics for the FD events might be larger or smaller than the mean. The histograms for the characteristics of all FD events during 1981 ~ 2022 are displayed in Fig.S5. It can be seen in Fig.S5 (a) ~ (c) that $FD_{RZSM}$ mostly onsets in 2 ~ 3 pentads and recovers in 2 ~ 5 pentads, with the duration$_{Total}$ of 4 ~ 7 pentads. All $FD_{SESR}$ and $FD_{MESR}$ events have similar duration$_{Onset}$, duration$_{Recovery}$, and duration$_{Total}$, with duration$_{Onset}$ mainly ranging between 2 and 4 pentads, duration$_{Recovery}$ between 2 and 7 pentads, and duration$_{Total}$ between 6 and 9 pentads."*

20. Line 321: The maps of severity, duration, and intensity of flash droughts detected by MESR are almost evenly distributed, especially compared to the other two methods. Although there are some concentrated areas, overall, these maps appear evenly distributed, suggesting that this method does not respond significantly to regional

characteristics, unlike the RZSM and SESR methods. Are there any reasons for this? If so, it would be helpful to discuss this further in the main text.

**Response:** Thank you for the comments. The spatial heterogeneity of the FD characteristics has been analyzed, combined with the land use types, Aridity Index (AI), and the ratio of mean annual ET and PET in the revised manuscript, which has been mentioned in Comment 2 (see lines 321-334, lines 361-364, lines 381-386, lines 398-405).

21. Lines 325-326: Are there any specific characteristics in the northern NCP that cause this change (e.g., land cover or background aridity)? It would be helpful if the authors could justify the spatial differences in the characteristics of droughts using this method.

**Response:** Thank you for the comments. The spatial heterogeneity of the FD characteristics in the original manuscript has been analyzed, combined with the land use types, Aridity Index (AI), and the ratio of mean annual ET and PET in the revised manuscript (see Comment 2 and lines 321-334, lines 361-364, lines 381-386, lines 398-405).

22. Line 361: A considerable number of flash droughts detected by MESR are categorized as grade 4, but they are not confirmed or detected by the other two methods. Moreover, this study relies on reanalysis ET, PET, and SM data, which contain uncertainties. Therefore, it is unclear whether the FD4 events detected by this method are actual flash drought events.

**Response:** Thank you for the comments. We have reanalyzed the characteristics of FD events based on two other reanalysis datasets, GLEAM and GLDAS 2 datasets, and obtained similar findings (see lines 437-442 and lines 569-574).

*"Figure S6 (a) displays the proportion of FD1 ~ 4. $FD_{RZSM}$ is primarily focused on $FD_{RZSM}3$, which is followed by $FD_{RZSM}2$ and $FD_{RZSM}4$, and finally, $FD_{RZSM}1$. On the other hand, $FD_{SESR}$ is concentrated on $FD_{SESR}1 ~ 2$, $FD_{SESR}3$, and $FD_{SESR}4$. $FD_{MESR}4$ accounts for the largest amount of $FD_{MESR}$, while the proportions of $FD_{MESR}3$, $FD_{MESR}2$, and $FD_{MESR}1$ steadily decrease. The FD1 ~ 4 proportions based on GLEAM and GLDAS 2 datasets in Fig.S6 (b) and (c) also exhibit a great similarity to those based on the ERA5-Land dataset in Fig.S6 (a)."*

[Figure]

**Figure S6 Proportion of FD1 ~ 4 for $FD_{RZSM}$, $FD_{SESR}$, and $FD_{MESR}$ based on (a) ERA5-Land, (b) GLEAM, and (c) GLDAS 2 datasets.**

*"Besides, the FD characteristics identified based on ERA5-Land, GLEAM, and GLDAS 2 datasets are displayed in Figs.S5, S13 and S14. The distributions of the $FD_{RZSM}$, $FD_{SESR}$, and $FD_{MESR}$ characteristics based on various datasets are comparable with the exception of the $FD_{MESR}$ intensity based on GLDAS 2. Meanwhile, the proportions of*

*various FD grades determined by intensity from diverse data sources also demonstrate an indisputable resemblance, as seen in Fig.S6. The similarity of pentad RZSM, SESR, and MESR percentiles from various datasets, as well as the FD characteristics based on various datasets, effectively demonstrates the reliability of our findings."*

23. Line 366: "Decrease trend" should be changed to "decreasing trend" for grammatical correctness.

**Response:** Thank you for the comments. We have revised the presentation (see lines 453-455).

*"The affected area of FD$_{RZSM}$ fluctuates between 0 and 60% with a clear decreasing trend, while that of FD$_{SESR}$ ranging from 0 to 80% gradually rises and FD$_{MESR}$ decreases ranging from 0 to 80%."*

24. Line 387: "Increases" should be changed to "increase" to match the plural subject ("duration and severity").

**Response:** Thank you for the comments. We have revised "increases" into "increase" (see lines 475-476).

*"Similarly, the duration$_{Total}$ and severity of FD$_{SESR}$ and FD$_{MESR}$ increase in the southern and west-central NCP, but decrease in the east-central NCP."*

25. Line 389: "Increases" should be changed to "increase" to match the plural subject ("duration and severity").

**Response:** Thank you for the comments. We have revised the presentation and corrected the "increases" in the revised manuscript.

26. Line 399: Are these drought events categorized as flash drought or they are conventional drought events?

**Response:** Thank you for the comments. We have researched again for well-documented FD events and used two typical FD events occurring in 2017 and 2019 to evaluate the applicability of three FD identification methods in Section 3.4 (see Comment 1; Section 3.4 in lines 481-502). The FD events in 2017 and 2019 are identified by soil moistures, and both are well recorded with the detailed development and evolution records in Xue (2023) and Yao et al. (2022), further confirmed in Chen et al. (2024) as well. Grids suffering FD$_{RZSM}$, FD$_{SESR}$, and FD$_{MESR}$ are identified in the revised manuscript and compared with the development and evolution records of the two real FD events, demonstrating good consistency.

27. Lines 395-495: I am not sure if this section can provide any reliable results, mainly because it is unclear whether those events are flash droughts or not, and if they are, how they were detected. Moreover, this study utilizes reanalysis data, which have inherent uncertainties. Therefore, this study could benefit from a comparison between these three methods and some real flash drought events in the study area that occurred recently and have reliable information on their characteristics.

**Response:** Thank you for the comments. We have researched again for welldocumented FD events, and used two typical FD events occurring in 2017 and 2019 to evaluate the applicability of three FD identification methods in Section 3.4, which has been mentioned in Comment 1 (see Section 3.4 in lines 481-502).

28. Figure 9: What is the meaning of 'The color bands represent the pentads with FD events from June to August of that year.' For example, in 1989 event, what does it mean to have 44 (max) FD events from June to August in each pixel?
**Response:** Thank you for the comments. We have modified the presentation of the typical FD events to the pentad evolution of FD events, where the color bands have been removed (see Figure 9 in lines 500-502).

29. Lines 418-419: Do you have any reference for this? What are the main characteristics of these regions, mainly in term of land cover?
**Response:** Thank you for the comments. We have added the reasons why these regions are the hotspots combined with the land use types, Aridity Index (AI), and the ratio of mean annual ET and PET (see lines 507-513).
*"These hotspots are situated in the northeastern and southwestern NCP, respectively. High hotspot indicator regions are likely to encounter frequent and severe FD events. Northern NCP has low AI and significant evaporative pressure. There is a significant water demand in the woodland. The eastern NCP is similar to the northern NCP with low AI and high evaporative pressure. Although it is cultivated land where irrigation might effectively alleviate the FD development, its coastal position promotes evaporation. In the southern NCP, the high temperature prolongs the FD duration$_{Total}$ and increases the evaporation loss despite of the high AI and low evaporative pressure, accelerating the decline of soil moisture and rendering it prone to FD events."*

30. Line 448: It would be better if the authors could start this section with a sentence stating that they developed MESR method in this study.
**Response:** Thank you for the comments. We have revised the presentation (see line 576).
*"This study developed a new FD identification method called MESR, which is a modified version of SESR."*

31. Line 476: change 'decrease' to 'decreasing trend'
**Response:** Thank you for the comments. We have corrected the "decrease" into "decreasing" (see lines 605-606).
*"The annual affected area of FD$_{RZSM}$ and FD$_{MESR}$ in this study exhibit a significant decreasing trend, but that of FD$_{SESR}$ exhibits a slow increase."*

32. Line 477: change 'slowly' to 'slow'
**Response:** Thank you for the comments. We have corrected the "slowly" into "slow" (see lines 605-606).
*"The annual affected areas of FD$_{RZSM}$ and FD$_{MESR}$ in this study exhibit a significant decreasing trend, but that of FD$_{SESR}$ exhibits a slow increase."*

33. Lines 489-490: The 50th percentile of SESR is not greater than zero in all regions. As shown in Figure S7, in the vast majority of regions, it is around zero or even lower. Moreover, in regions with a higher evaporation baseline, the 50th percentile is slightly higher than zero, but this is not a disadvantage of the SESR method. Perhaps one reason your method shows spatially heterogeneous frequencies is this, which could lead to missing some rapidly developing events, particularly in wetter climate regimes.

**Response:** Thank you for the comments. Section 4.4 illustrates the rationality of the spatially heterogeneous frequencies between $FD_{SESR}$ and $FD_{MESR}$. The spatial distribution of mean $SESR_{50th}$ and $\Delta SESR_{50th}$ in Fig.S16 on the $FD_{SESR}$ pentads determined that there might be an overestimation in the $FD_{SESR}$ frequency. The distributions of average $SESR_{40th}$ and $\Delta SESR_{40th}$ further show that the unobserved $FD_{SESR}$ events would be captured in the northern NCP. Figure S17 displays the frequency of $SESR_{50th}$, $\Delta SESR_{50th}$, $SESR_{40th}$, and $\Delta SESR_{40th}$ when $FD_{SESR}$ occurs, supporting the likelihood of $FD_{SESR}$ frequency overestimation in the NCP as well. The difference between $FD_{SESR}$ and $FD_{MESR}$ identification results can be traced back to two aspects: PDFs fitting and variable thresholds. The PDFs fitting decreases the FD frequency while the variable threshold increases that. The contribution of PDFs fitting is less than that of variable thresholds in the northeastern and west-central NCP but greater in the other regions. Therefore, the frequency difference between $FD_{SESR}$ and $FD_{MESR}$ is mainly due to the variable thresholds in the northeastern and west-central NCP, and due to the PDFs fitting in the other regions. The relative contributions of PDFs fitting and variable thresholds demonstrate the rationality of the FD frequency difference between $FD_{SESR}$ and $FD_{MESR}$ (see lines 648-691).

*"**4.4 Attribution analysis of the frequency difference between $FD_{SESR}$ and $FD_{MESR}$***

*Compared to $FD_{SESR}$, the $FD_{MESR}$ frequency shows spatial heterogeneity, which is connected to the frequency distribution of $SESR_{50th}$ and $\Delta SESR_{50th}$. Figure S16 illustrates the spatial distribution of mean $SESR_{50th}$ and $\Delta SESR_{50th}$ on the $FD_{SESR}$ pentads. Except for the southern NCP where the $SESR_{50th}$ is less than 0, the $SESR_{50th}$ in the northern and central NCP is mainly more than 0. Besides, the $\Delta SESR_{50th}$ in NCP is all larger than 0, which makes it possible that $SESR_{40th}$ cannot indicate the real low evaporative stress value, as well as the underestimation of evaporative stress value and inaccurate capture of $FD_{SESR}$ events that would not occur. Therefore, it may result in the overestimation in the NCP (see Sect.3.2). Theoretically, the $SESR_{40th}$ and $\Delta SESR_{40th}$ in the NCP should be smaller than 0, representing the low evaporative stress value and decreasing ESR. The distributions of average $SESR_{40th}$ and $\Delta SESR_{40th}$ in Fig.S16 (c) (d) indicate that both are less than 0 except for the northern NCP, where the $SESR_{40th}$ is greater than 0. It further shows that the unobserved $FD_{SESR}$ events would be captured in the northern NCP. The frequency of $SESR_{50th}$, $\Delta SESR_{50th}$, $SESR_{40th}$, and $\Delta SESR_{40th}$ when $FD_{SESR}$ occurs are determined in Fig.S17. The $SESR_{50th}$ and $\Delta SESR_{50th}$ values are mostly greater than 0, particularly the $\Delta SESR_{50th}$, corresponding to Fig.S16 (a) (b). However, most $SESR_{40th}$ and $\Delta SESR_{40th}$ are below 0. It could not be ignored that both $SESR_{40th}$ and $\Delta SESR_{40th}$ greater than 0 account for around 35%, supporting the likelihood of $FD_{SESR}$ overestimation in the NCP.*

*The FD identification based on SESR and MESR differs in that the ESR is standardized by its mean to construct MESR rather than being normalized into SESR, MESR is fitted with various PDFs rather than EDF, and the variable thresholds are utilized in the $FD_{MESR}$ identification. MESR and SESR are both linearly converted from ESR in order to facilitate comparison of the FD identification results between different regions. Converting ESR to MESR rather than SESR only makes the ESR standardization process more reasonable, but the linear translation of ESR into SESR or MESR has no effect on its corresponding percentile. Therefore, the difference between $FD_{SESR}$ and $FD_{MESR}$ identification results can be traced back to two aspects: PDFs fitting and variable thresholds.*

*To demonstrate their contribution to the difference between $FD_{SESR}$ and $FD_{MESR}$, the thresholds in $FD_{SESR}$ method are referred to. The fixed thresholds, MESRonset1 = 40, MESRonset2 = 20, ∆MESRonset = 40, and MaxMESRchange = 25, are applied in the $FD_{MESR}$ method, which is called $FD_{MESR-invariable}$. Figure S18 (a) ~ (c) displays the frequency of $FD_{MESR-invariable}$ and the differences between $FD_{MESR-invariable}$ frequency and $FD_{SESR}$ and $FD_{MESR}$ frequency. Figure S18 (d) (e) also show the contribution of PDFs fitting and variable thresholds to the differences between $FD_{SESR}$ and $FD_{MESR}$ frequency, respectively, as well as the relative contribution in Fig.S18 (f). $FD_{MESR-invariable}$ and $FD_{MESR}$ have a comparable frequency spatial distribution, with higher frequency in the north-central and lower in the south of NCP. Besides, $FD_{MESR-invariable}$ has a lower frequency than $FD_{SESR}$ and $FD_{MESR}$ in NCP, indicating that the PDFs fitting decreases the FD frequency while the variable thresholds increase that. According to Fig.S18 (d) (e), the PDFs fitting has a negative impact on the difference between $FD_{SESR}$ and $FD_{MESR}$ frequency in the northeastern and west-central NCP, whilst the variable thresholds have a positive impact. However, it is opposite in the other NCP regions. As shown in Fig.S18 (f), the relative contribution of the variable thresholds to the PDFs fitting mostly ranges between -2 and -1 in the northeastern and west-central NCP, but between -1 and 0 in the other regions. Therefore, the absolute values of the relative contributions of the PDFs fitting are larger than 1 in the northeastern and west-central NCP, but less than 1 in the other regions. Taking into account the negative contribution of PDFs fitting in the NCP, it can be assumed that the contribution of PDFs fitting is less than that of variable thresholds in the northeastern and west-central NCP but greater in the other regions. Therefore, the frequency difference between $FD_{SESR}$ and $FD_{MESR}$ frequency is mainly due to the variable thresholds in the northeastern and west-central NCP, and due to the PDFs fitting in the other regions.*

*The $FD_{SESR}$ frequency is lower than the $FD_{MESR}$ frequency in the northeastern and west-central NCP while larger in the other regions. Although the $FD_{SESR}$ frequency is overestimated in the NCP, particularly the northern NCP, PDFs fitting decreases the FD frequency in the $FD_{MESR}$. But the variable thresholds increasing the FD frequency contribute more in the northeastern and west-central NCP, resulting in a greater $FD_{MESR}$ frequency than $FD_{SESR}$ frequency. In the other regions where PDFs fitting plays a larger role, the decreasing from PDFs fitting surpasses the increasing from the variable thresholds, making that the $FD_{MESR}$ frequency is lower than the $FD_{SESR}$ frequency."*

[Figure]

*Figure S16 Distribution of the mean (a) SESR$_{50th}$, (b) ΔSESR$_{50th}$, (c) SESR$_{40th}$, and (d) ΔSESR$_{40th}$ on FD$_{SESR}$ pentads.*

[Figure]

*Figure S17 Histogram of (a) SESR$_{50th}$, (b) ΔSESR$_{50th}$, (c) SESR$_{40th}$, and (d) ΔSESR$_{40th}$ on FD$_{SESR}$ pentads.*

[Figure]

*Figure S18 Frequency of FD$_{MESR-invariable}$ and its difference with FD$_{MESR}$ and FD$_{SESR}$.*

34. Lines 498-499: Actually, there are three differences. In addition to what you have mentioned (PDF fitting and threshold), your method multiplies ESR by its mean, whereas in the original SESR, ESR anomalies are standardized. Please include this in the main text.

**Response:** Thank you for the comments. The three differences between FD identification using MESR and SESR have been supplemented in lines 662-668. Due to that the linear transformation conducted on ESR does not influence the percentile results, which only increases the rationality of the ESR standardization process, the difference between $FD_{SESR}$ and $FD_{MESR}$ identification results can be traced back to two aspects: PDFs fitting and variable thresholds.

*"The FD identification based on SESR and MESR differs in that the ESR is standardized by its mean to construct MESR rather than being normalized into SESR, MESR is fitted with various PDFs rather than EDF, and the variable thresholds are utilized in the $FD_{MESR}$ identification. MESR and SESR are both linearly converted from ESR in order to facilitate comparison of the FD identification results between different regions. Converting ESR to MESR rather than SESR only makes the ESR standardization process more reasonable, but the linear translation of ESR into SESR or MESR has no effect on its corresponding percentile. Therefore, the difference between $FD_{SESR}$ and $FD_{MESR}$ identification results can be traced back to two aspects: PDFs fitting and variable thresholds."*

35. Line 528: Again, these historical events are not reliable, mainly because as the authors mentioned that there is no detailed information on these events, and it is unclear whether they were flash droughts or not. The combination of uncertainty in the reanalysis datasets used in this study and the lack of adequate information on these events cannot lead to a reliable conclusion.

**Response:** Thank you for the comments. We have researched again for well-documented FD events, and used two typical FD events occurring in 2017 and 2019 to evaluate the applicability of three FD identification methods in Section 3.4 (see Comment 1; Section 3.4 in lines 481-502 and lines 707-708). Besides, the uncertainties from the different reanalysis datasets have been evaluated in the Section 4.1 of the revised manuscript (see Comment 9 and lines 552-574).

*"Meanwhile, this study demonstrated the three FD methods could accurately identify the true FD events through the two typical historical events in Sect.3.4."*

36. Lines 447-448: Please rephrase this sentence as follows: "There are notable differences in the spatial distribution of severity among the three FD methods."

**Response:** Thank you for the comments. We have rephrased this sentence (see lines 729-730).

*"There are notable differences in the spatial distribution of severity among the three FD methods."*

37. Lines 545-549: This study could benefit from a deeper discussion on the main

reasons for changes in the frequencies and characteristics of FD events. Such a discussion could incorporate land cover or the background aridity of the study area.

**Response:** Thank you for the comments. The spatial heterogeneity of the FD characteristics in the original manuscript has been analyzed, combined with the land use types, Aridity Index (AI), and the ratio of mean annual ET and PET in the revised manuscript, which has been mentioned in Comment 2 (see lines 321-334, lines 361-364, lines 381-386, lines 398-405).

38. Lines 561-565: Isn't it obvious? As the depth of soil moisture increases, the impact of flash droughts on SM can become less pronounced. Moreover, this study used reanalysis SM data, so finding these trends in reanalysis datasets is not surprising!

**Response:** Thank you for the comments. We have identified the FD events based on GLEAM and GLDAS 2 datasets as well, and obtained similar findings (see lines 522-523). Moreover, the mean impacts of the thresholds for each unit on the FD frequency based on different datasets are shown in Table S3, which further measures the impacts of thresholds on the FD frequency. Overall, the minimum FD duration, as well as the continuous pentads that FD indexes such as RZSM, ΔSESR, and ΔMESR should exceed the FD termination thresholds, is sensitive to the FD identification, while the thresholds for the decreasing indexes in the FD onset stage are insensitive (see lines 536-546). The similar effects of various thresholds on the frequency based on various datasets increases the credibility of the findings (see lines 715-718).

*"The mean $FD_{RZSM}$ frequency based on various soil layers from GLEAM and GLDAS 2 datasets are listed in Table S2, where the FD frequency follows the same pattern."*

*"To further demonstrate the impacts of thresholds on the FD frequency, the mean impacts of the thresholds for each unit on the FD frequency are shown in Table S3, corresponding to the FD frequency in Figs.11, S9, and S10. For $FD_{RZSM}$, the impacts of thresholds on the mean frequency are ranked as follows: "RZSMtermination" < "RZSMpentad1" < "RZSMonset1" < "RZSMonset2" < "MinDuration" < "RZSMpentad2". For $FD_{SESR}$, the impacts of thresholds on the FD frequency ranging from small to large are: "ΔSESRonset" < "SESRonset1" < "SESRonset2" < "MaxSESRchange" < "SESRpentad" < "MinDuration". For $FD_{MESR}$, the impacts of thresholds are: "ΔMESRonset" < "MESRonset1" < "MESRonset2" < "MaxMESRchange" < "MESRpentad" < "MinDuration". Therefore, the minimum FD duration ("MinDuration"), as well as the continuous pentads that FD indexes such as RZSM, ΔSESR, and ΔMESR should exceed the FD termination thresholds ("RZSMpentad2", "SESRpentad", and "MESRpentad"), is sensitive to the FD identification, while the thresholds for the decreasing indexes in the FD onset stage ("RZSMtermination", "RZSMpentad1", "RZSMonset1", "ΔSESRonset", "SESRonset1", "SESRonset2", "ΔMESRonset", "MESRonset1", and "MESRonset2") are insensitive."*

*"The sensitivity of thresholds to the $FD_{RZSM}$, $FD_{SESR}$ and $FD_{MESR}$ frequency are measured, as well as that based on various datasets. Table S3 demonstrates that the rankings of the effects of various thresholds on the mean frequency based on various datasets exhibit notable similarities, increasing the credibility of the findings."*

**Table S3 The mean frequency change of various FD events caused by one unit**

*change in the FD identification thresholds*

| FD identification method | Parameter | ERA5-Land | GLEAM | GLDAS 2 |
|---|---|---|---|---|
| $FD_{RZSM}$ | RZSMpentad1 | 0.0031 | 0.0024 | 0.0025 |
| | RZSMonset1 | -0.0121 | -0.0167 | -0.0075 |
| | RZSMonset2 | 0.0227 | 0.0297 | 0.0137 |
| | RZSMpentad2 | -0.0788 | -0.1493 | -0.0411 |
| | RZSMtermination | -0.0022 | -0.0005 | -0.0014 |
| | MinDuration | -0.0387 | -0.0857 | -0.0179 |
| $FD_{SESR}$ | Minduration | -0.6137 | -0.1468 | -0.1043 |
| | SESRonset1 | 0.0022 | -0.0018 | -0.0002 |
| | SESRonset2 | 0.0277 | 0.0094 | 0.0108 |
| | $\Delta$SESRonset | 0.0007 | 0.0016 | 0.0010 |
| | SESRpentad | -0.0978 | -0.0299 | -0.0196 |
| | MaxSESRchange | 0.0457 | 0.0161 | 0.0136 |
| $FD_{MESR}$ | Minduration | -0.5410 | -0.2507 | -0.1126 |
| | MESRonset1 | 0.0021 | 0.0018 | -0.0007 |
| | MESRonset2 | 0.0249 | 0.0164 | 0.0131 |
| | $\Delta$MESRonset | -0.0009 | 0.0090 | 0.0040 |
| | MESRpentad | -0.1260 | -0.0054 | -0.0122 |
| | MaxMESRchange | 0.0476 | 0.0096 | 0.0075 |

---

## Author Comment (AC3)

Dear editor:

On behalf of all the contributing authors, I would like to express our sincere appreciations of your letter and the constructive comments from Referee #2 concerning our article entitled "Flash drought characteristics based on three identification methods in the North China Plain, China". All the comments are very helpful for revising our paper. We have studied and discussed all the comments point-by-point carefully, and accordingly made substantial revisions to our paper. All the changes we have made were in the red-colored text. Our point-by-point responses to all the comments are provided below in the blue-colored texts.

\*\*\*\*\*\*\*\*\*\*\*\*\*\*\*\*\*\*\*\*\*\*\*\*\*\*\*\*\*\*\*\*\*\*\*\*\*\*\*\*\*\*\*\*\*\*\*\*\*\*\*\*\*\*\*\*\*\*\*\*\*\*\*\*\*\*\*\*\*

Zhang et al, have studied flash drought characteristics in China based on three indicators. This manuscript does not meet the requirements for publication in the HESS journal.

1. First of all, the manuscript needs significant improvements in terms of language, there are so many grammatical errors, long sentences that make it hard to follow the text, and so many abbreviations even in the Abstract that confuse the reader.

**Response:** Thank you for the comments. We apologize for the poor language of our manuscript. We have corrected the grammar errors, revised the long sentences into short sentences, and made efforts to reduce the abbreviations in the Abstract. We have now worked on both language and readability and have also involved native English speakers for language corrections. We really hope that the flow and language level have been substantially improved.

[Figure]

2. In the Introduction, the literature review merely mentions other studies using different drought indicators (~3 paragraphs and introducing a new abbreviation in every line) without highlighting the knowledge gap and the challenges that need to be addressed.

**Response:** Thank you for the comments. The literature review in the Introduction introduces many drought indicators but lacks focus and knowledge gaps. We have revised the writing logic of the Introduction. Firstly, it pointed out the serious harm of FD and the widespread global concern. Then, the application of FD identification indicators, which are developed by conventional drought indicators, soil moisture, and atmospheric evaporation demand, was introduced. The advantages and disadvantages of these FD identification indicators were pointed out, emphasizing two widely applied

indicators, RZSM and SESR. After that, previous applications of RZSM and SESR for FD identification were discussed, especially their findings on FD characteristics. It was also pointed out that FDs based on RZSM and SESR are from agricultural and meteorological perspectives, respectively. The significance of exploring FD from different perspectives was highlighted. Finally, limitations of FD identification by SESR were displayed. In the last paragraph of Introduction, improvement methods to address those limitations were proposed, and the objectives of this paper were stated (see Introduction in lines 25-109).

**1 Introduction**
*The terrestrial water cycle has accelerated and droughts have become more frequent under global warming. Persistent and slow drought disasters, caused by long-term inadequate precipitation, can severely harm ecology, agriculture, and economy (Li et al., 2020; Limones, 2021). In addition to these long-term and slow droughts, some droughts characterized by rapid occurrence and intensification, known as flash droughts (FDs), also occur frequently (Deng et al., 2022; Yuan et al., 2023). FDs are driven by both water and energy limitations, and their rapid onset poses significant challenges for drought monitoring and forecasting (Yuan et al., 2023; Zhang et al., 2022a). A severe FD occurred in the United States during the summer of 2012. The conditions rapidly shifted from no drought to extreme drought within less than a month. Its rapid intensification made prediction difficult, resulting in economic losses exceeding $30 billion (Yuan et al., 2023). Other notable FD events included those in western Russia in 2010, southern Great Plains in 2015, and southern China in 2019 (Edris et al., 2023; Hunt et al., 2021; Wang and Yuan, 2021). Because of the severe societal and environmental impacts of FD, its occurrence and development have received significant attention worldwide (Deng et al., 2022; Li et al., 2020).*

*FDs frequently occur alongside increasing temperature and decreasing precipitation, resulting in a rapid increase in evapotranspiration and a reduction in soil moisture (Tyagi et al., 2022). Consequently, some hydrological and meteorological variables have been applied as key indicators to identify FD in previous studies. Currently, FD identification methods can be categorized into three main classifications on the basis of conventional drought indicators, soil moisture, and atmospheric evaporation demand.*

*The conventional drought indicators employed for identifying FDs include United States Drought Monitor (USDM) (Chen et al., 2019; Edris et al., 2023; Otkin et al., 2019; Pendergrass et al., 2020), standardized precipitation evapotranspiration index (SPEI) (Fu and Wang, 2022; Noguera et al., 2020; Noguera et al., 2021), and standardized precipitation index (SPI) (Noguera et al., 2021; Parker et al., 2021). However, these conventional drought indices have drawbacks. For example, USDM provides weekly updates on drought location and intensity but is applicable only within the United States. SPI focuses only on the precipitation deficit and neglects the influences of temperature and evaporation. SPEI integrates both insufficient precipitation and increased evapotranspiration, yet its response to climatic changes is slow (Deng et al., 2022). Consequently, several specific indicators for FD identification that respond quickly to climate change and are applicable in broader regions have also been developed using soil moisture and climate factors.*

*Soil moisture reflects the combined influence of precipitation and evapotranspiration and is directly related to agricultural production. Therefore, soil moisture-based indicators have been widely employed in agricultural flash drought identification. These indices include root zone soil moisture (RZSM, Yuan et al., 2019), soil moisture*

*index (SMI, Hunt et al., 2009, 2021), soil moisture volatility index (SMVI, Osman et al., 2021; Osman et al., 2022), and flash drought stress index (FDSI, Sehgal et al., 2021). SMI uses wilting and field capacity soil metrics to analyze the changes in soil moisture stress. However, these metrics are influenced by soil types and can be challenging to estimate (Garg et al., 2017; Otkin et al., 2018; Rab et al., 2011; Sehgal et al., 2021). SMVI must be reset by precipitation and is sensitive to interruptions associated with drought onset (Osman et al., 2021). Due to the limited research and application of SMVI, its effectiveness and universality need further verification. In addition, FDSI combines soil moisture and climate factors, which can comprehensively reflect FD stress. However, the diverse required data and complex calculations limit its widespread application (Sehgal et al., 2021). RZSM can directly monitor the rapid decreases in soil moisture over a short period and is closely related to soil moisture stress (Yuan et al., 2019). Owing to its clear physical mechanism, simple data requirements, and ease of calculation, RZSM is widely used worldwide.*

*In addition to soil moisture, various indicators have been employed to identify FDs on the basis of the atmospheric evaporation demand. The evaporative stress index (ESI) is calculated as 1 minus the ratio of actual evapotranspiration (ET) to potential evapotranspiration (PET) (Ahmad et al., 2022; Anderson et al., 2007; Hunt et al., 2021; Otkin et al.,2019). It is sensitive to energy- and water-limited situations, although the spatial resolution is typically 5 ~ 10 km (Anderson et al., 2007). Standardized evapotranspiration deficit index (SEDI; Vicente-Serrano et al., 2010) evaluates both heatwave- and water deficit-driven FDs by measuring the difference between ET and PET. However, SEDI exhibits significant uncertainty in humid and sub-humid regions (Li et al., 2020; Rakkasagi et al., 2023). The evaporative demand drought index (EDDI; Hobbins et al., 2016) detects FDs through the PET response to surface dryness anomalies. While it provides weekly water stress, it can be utilized only in the contiguous United States (Rakkasagi et al., 2023). Standardized evaporative stress ratio (SESR; Christian et al., 2019) incorporates near-surface state variables and plant health, making it sensitive to rapid changes in evaporative stress and capable of capturing early FD signals (Gou et al., 2022). Identifying FDs by SESR is based on the objective percentiles on a grid, making it globally applicable (Christian et al., 2019).*

*Therefore, RZSM and SESR are commonly utilized as indicators for identifying agricultural FDs and meteorological FDs in regional studies. For example, Yuan et al. (2019) characterized FD frequency and duration using RZSM and projected FD severity and risk exposure. They reported that southern China experiences more frequent, severe, and higher-risk FDs. Zhang et al. (2022a) assessed FDs using RZSM and simulated FD intensity by three machine learning models, revealing greater FD intensity in southeastern China. Zhong et al. (2022) evaluated FD spatial and temporal characteristics, such as duration, severity, and intensity, via SESR and identified increasing FD risk from west to east in the Pearl River Basin, China. Gou et al. (2022) illustrated the spatial distribution of FD frequency and duration using SESR and tracked FD spatial paths in the Huaibei Plain of China from 2001 to 2019. Deng et al. (2022) used SESR to determine global FD spatiotemporal characteristics and found a significant decrease in FD coverage from 1981 to 2020. Mukherjee and Mishra (2022b) investigated global FD frequency and intensity using SESR and RZSM, highlighting the influence of climate factors and background aridity on FD onset and evolution. Therefore, RZSM and SESR are widely applied to identify regional FD (i.e., $FD_{RZSM}$ and $FD_{SESR}$). $FD_{SESR}$, identified by anomalous evaporative stress, is classified as a meteorological drought. However, $FD_{RZSM}$, determined by soil moisture, is categorized as an agricultural drought. Although they are interrelated, they identify FDs from*

*different perspectives. Exploring FD characteristics from multiple perspectives plays an important role in the comprehensive understanding and effective response to FD events. Therefore, this study employs RZSM and SESR to further analyze regional FD characteristics and investigate the effects of climate factors and background aridity. However, the SESR application has several problems. When a FD identified via SESR, both SESR and the change in SESR ($\Delta SESR$) are assumed to follow normal distributions. Therefore, the 50th percentiles of $\Delta SESR$ ($\Delta SESR_{50th}$) equal 0, indicating no change in SESR ($\Delta SESR = 0$). A $\Delta SESR$ below the 40th percentile of $\Delta SESR$ ($\Delta SESR_{40th}$) denotes a decrease in the SESR. However, whether SESR and $\Delta SESR$ follow normal distributions remains uncertain. If they do not, $\Delta SESR_{40th}$ may not be less than 0. In the study by Gou et al. (2022), the 36th percentile of $\Delta SESR$ ($\Delta SESR_{36th}$) corresponded to an increase in SESR, where $\Delta SESR_{36th}$ was greater than 0. This phenomenon might be because SESR gradually decreases during dry periods and increases during precipitation process. Consequently, $\Delta SESR$ can be less than 0 during dry periods and greater than 0 during precipitation periods, potentially leading to underestimation or overestimation of FDs. Furthermore, Christian et al. (2019) did not provide a criterion for SESR at FD onset, only specifying a criterion for the minimum SESR at the FD onset stage greater than the 20th percentile ($SESR_{20th}$). This leads to the possibility that SESR at FD onset is below $SESR_{20th}$. Therefore, it is necessary to add a criterion for SESR at FD onset.*

*In this study, a new method based on the multiples of the mean evaporative stress ratio (MESR) for FD identification was developed to address the aforementioned limitations. Unlike SESR, MESR does not rely on the assumption of a specific probability density function (PDF) for evaporative stress ratio (ESR). Instead, MESR and $\Delta MESR$ are fitted by multiple PDFs and converted into percentiles. Variable thresholds are employed for identifying FDs to guarantee that the thresholds of MESR and $\Delta MESR$ are less than 0, reflecting a lower level of MESR and a decrease in MESR. The objectives of this study are to: (1) propose an improved FD identification method called MESR based on SESR; (2) characterize FDs in the North China Plain (NCP) from 1981 to 2022 using RZSM, SESR, and MESR and investigate FD temporal and spatial trends; and (3) identify FD hotspots in the NCP and evaluate the impact of thresholds on FD identification."*

3. The same happens in the Discussion where a lot of other studies are mentioned without explaining what has been done differently or what is different in their results from those other studies.

**Response:** Thank you for the comments. We have revised Section 4.1, which compared FD characteristics of this study with those of previous studies. The revised manuscript has emphasized the consistency between the findings of this study and those of others and explained the reason for the inconsistency with other studies. In addition, the impact of FD identification indicators on FD characteristics has also been discussed (see Section 4.1 in lines 554-606). Furthermore, the innovation of this study, namely what has been done differently, has also been added in Section 4.4 (see lines 673-685).

lines 554-606
*"**4.1 FD characteristics compared with previous studies**
A new FD identification method called MESR, which is an improved version of SESR, was developed in this study. On the basis of RZSM, SESR, and MESR, FD events in the NCP from 1981 to 2022 were identified from both agricultural and meteorological*

*drought perspectives. The spatial distributions of FD characteristics, such as frequency, duration, severity, and intensity, were analyzed. These findings suggest that FD frequency is high in the north-central NCP and low in the southern NCP. Frequency of agricultural FD ($FD_{RZSM}$) ranges mostly 0 ~ 0.5 times/year, whereas $FD_{RZSM}$ IR ranges primarily from 10 to 50 %. Previous studies have also reported $FD_{RZSM}$ frequency in the NCP. Yuan et al. (2019) estimated $FD_{RZSM}$ frequency in the NCP at 0 ~ 0.6 times/year using the fifth Coupled Model Intercomparison Project (CMIP5). Mukherjee and Mishra (2022a, 2022b) identified $FD_{RZSM}$ events from 1980 to 2018 using ERA5, reporting a $FD_{RZSM}$ frequency of 0.08 ~ 0.38 times/year and a $FD_{RZSM}$ IR of 15 ~ 30 % in the NCP. Wang and Yuan (2022) found a $FD_{RZSM}$ frequency of primarily 0.1 ~ 0.4 times/year in the NCP by ERA5, and Mahto and Mishra (2023) showed a $FD_{RZSM}$ IR of 11 ~ 15 % in the NCP using ERA5. These studies illustrate that frequency and IR determined by the ERA5 dataset are lower than those found in this study, which might be related to the different data sources used.*

*Moreover, frequency and IR of meteorological FDs ($FD_{SESR}$ and $FD_{MESR}$) identified in this study are mostly 0.4 ~ 0.8 times/year and 30 ~ 60%, respectively. Gou et al. (2022) identified $FD_{SESR}$ events in the Huaibei Plain (south of the NCP) from 2001 to 2019 using MODIS Global Evapotranspiration Project (MOD16) data with a spatial resolution of 500 m. They discovered a $FD_{SESR}$ frequency of 0.79 ~ 1.05 times/year, which was higher than that in this study. The Huaibei Plain is located in the sub-humid and humid regions with adequate precipitation, abundant heat, and developed agriculture, contributing to its higher $FD_{SESR}$ frequency than the NCP. Deng et al. (2022) found a $FD_{SESR}$ IR of 15 ~ 40 % from 1981 to 2020 in the NCP using ERA5-Land, whereas Christian et al. (2023) obtained a mean $FD_{SESR}$ IR of 25 ~ 35% in the NCP using four reanalysis datasets from 1980 to 2014. These variations highlight the influences of data sources and their spatial resolution on the FD identification. Overall, the FD frequency in this study is within a reasonable range compared with those in previous studies.*

*Agricultural FDs ($FD_{RZSM}$) are less frequent than meteorological FDs ($FD_{SESR}$ and $FD_{MESR}$). The $FD_{RZSM}$ identification is based on soil moisture from an agricultural perspective, whereas $FD_{SESR}$ and $FD_{MESR}$ focus on atmospheric evaporation demand from a meteorological perspective. $FD_{RZSM}$ and $FD_{SESR}/FD_{MESR}$ are independent, whereas $FD_{MESR}$ is an advanced version of $FD_{SESR}$. The higher $FD_{SESR}$ and $FD_{MESR}$ frequencies than $FD_{RZSM}$ frequency is related to the fact that meteorological FDs are directly determined by atmospheric conditions and is more sensitive to FD occurrence and development. In contrast, agricultural FDs are dependent on soil moisture and influenced by crop regulatory effects and human activities (Meng et al., 2024).*

*The onset and recovery stages of FD events were distinguished in this study. $FD_{RZSM}$ events have duration$_{Onset}$ of 2 ~ 4 pentads, duration$_{Recovery}$ of 3 ~ 6 pentads, and duration$_{Total}$ of 5 ~ 9 pentads. $FD_{SESR}$ and $FD_{MESR}$ have similar duration$_{Onset}$ (2 ~ 4 pentads), duration$_{Recovery}$ (4 ~ 6 pentads), and duration$_{Total}$ (7 ~ 9 pentads). The findings of this study align with those of other studies on the $FD_{RZSM}$ duration of the NCP. Yuan et al. (2019) determined $FD_{RZSM}$ duration$_{Total}$ to be 4 ~ 8 pentads in the NCP using CMIP5. They also found duration$_{Total}$ to be 6 ~ 8 pentads from 1951 to 2014 in the NCP with three reanalysis datasets with a 1° resolution (Yuan et al., 2023). Zhang et al. (2020) identified FD using RZSM using ERA5 with a 0.25° resolution from 2003 to 2018 and found $FD_{RZSM}$ duration$_{Onset}$ of 3 ~ 4 pentads, duration$_{Recovery}$ of 1 ~ 6 pentads, and duration$_{Total}$ of 5 ~ 10 pentads in the NCP. These findings show that different datasets within various spatial resolutions have a smaller effect on FD duration. Some studies have also identified $FD_{SESR}$ events. Deng et al. (2022) revealed $FD_{SESR}$*

*duration$_{Total}$ to be 6 ~ 9 pentads in the NCP by three reanalysis datasets. In the Huaibei Plain, the adjacent region of the NCP, Gou et al. (2022) found FD$_{SESR}$ duration$_{Onset}$ to be 3 ~ 4 pentads and duration$_{Recovery}$ to be 2 ~ 4 pentads. The FD$_{SESR}$ duration$_{Onset}$ in the NCP is high in the south but small in the north, but duration$_{Recovery}$ is high in the north, as shown in Figure S2. The Huaibei Plain is located in the south of the NCP, therefore duration$_{Onset}$ in the Huaibei Plain is greater, but duration$_{Recovery}$ is smaller than in the NCP. The findings of this study demonstrate significant rationality compared with previous findings.*

*The spatiotemporal variabilities in FD characteristics are explored in this study, revealing a decreasing trend in the NCP. The spatial distribution patterns of duration$_{Total}$ and severity trends are comparable. Similar trends have been shown in other studies. Chen et al. (2019) reported a decreasing FD affected area in the United States from 2000 to 2017 using USDM. Zhang et al. (2021) determined a decreasing tendency in FD$_{RZSM}$ frequency, duration, severity, and intensity in the Gan River Basin from 1961 to 2018 using the variable infiltration capacity (VIC) model driven by meteorological data. Liu et al. (2020) and Deng et al. (2022) also reported that there was a notable decline in FD$_{RZSM}$ and FD$_{SESR}$ affected area in the NCP. Besides, Christian et al. (2023) revealed a noteworthy reduction in FD$_{SESR}$ frequency in the NCP from 1950 to 2015 using four reanalysis datasets. However, other studies have indicated different trends that cannot be disregarded (Yuan et al., 2019, 2023; Zhang et al., 2022a; Zhang et al., 2022c), which concern study areas, data sources, and study periods (Liu et al., 2020a). In addition, FD tendency might be affected by FD identification methods as well. For example, Noguera et al. (2021) revealed that FD frequency identified by the SPI decreased in Spain, whereas those identified by the EDDI and SPEI increased slowly."*

lines 673-685

*"**4.4 Innovation and prospects**
This study introduced a new FD indicator, MESR, as an improvement over SESR for regional FD identification. FD identified by MESR addressed the limitations of SESR, and the detailed improvement was explained in Section 2.3.3. Multiple indicators (i.e., RZSM, SESR, and MESR) were utilized to identify and characterize FD events from both agricultural and meteorological perspectives in this study, providing a comprehensive investigation of FD events in the NCP. The differences in FD characteristics are related to background aridity and climatic conditions. Besides, a FD hotspot identification method that combines agricultural and meteorological FDs and considers FD frequency, severity, and intensity was proposed. Percentile was also introduced, and two hotspots that were prone to frequent, severe, and intense FD were identified. Furthermore, the impacts of different thresholds in FD identification methods on FD frequency were quantified. Theoretical and statistical analyses were conducted to demonstrate the improvement in MESR over SESR. Furthermore, control variables were used to investigate the rationality of the spatial heterogeneity in FD$_{MESR}$ and FD$_{SESR}$ frequencies. Typical historical FD events were analyzed to demonstrate the applicability of FD indicators in the NCP as well. However, it is still worth exploring in depth how to quantitatively evaluate whether SESR or MESR is the optimal FD indicator."*

4. The novelty of this work is questionable. They have used three indicators to identify flash drought frequency, intensity, severity, and duration of flash droughts. Two of these

indicators have been used before, and it is not well described why the third indicator might have outperformed the others. There are differences between the values derived from the indicators although the spatial pattern is similar. My concern is how one could say which indicator is reporting flash drought characteristics closest to reality. In lines 245-255, the authors have tried to justify the preference of MESR over SESR by showing their distribution, but this does not convince the reader, how about RZSM? It is hard to understand what makes this study different from other studies on flash drought identification in China!

**Response:** Thank you for the comments. We have emphasized the limitations of $FD_{SESR}$ in lines 91-98 and added the detailed reasons for using MESR instead of SESR to identify FD in lines 204-214. $FD_{MESR}$ is an improvement method proposed based on the limitations of $FD_{SESR}$. The differences between $FD_{MESR}$ and $FD_{SESR}$ and detailed improvement have also been applied in lines 102-106 and lines 623-625. The distributions of SESR and MESR have been illustrated in Section 3.1, emphasizing the necessity of improving $FD_{SESR}$ method and the improvement effect of $FD_{MESR}$ method (see Section 3.1 in lines 294-317). Furthermore, the limitations of $FD_{SESR}$ and the rationality of the spatially heterogeneous frequencies between $FD_{SESR}$ and $FD_{MESR}$ have been displayed in Section 4.2 (see Section 4.2 in lines 607-653).

Two typical FD events occurring in 2017 and 2019 were utilized to validate the applicability of three FD identification methods in Section 3.4 (see Section 3.4 in lines 477-499). Furthermore, the performance of MESR has been evaluated in Text S2 (see Text S2 in lines 23-55 of Supplementary Materials). Especially in Fig.S18 (c), MESR has slightly higher explanatory ability for RZSM than SESR (see lines 42-44 of Supplementary Materials). Based on the maximal information coefficient (MIC), the correlation between pentad SPI series and MESR and SESR percentile series from 1981 to 2022 in the NCP have also been explored. A stronger correlation between SPI and MESR than between SPI and SESR has been shown in Fig.S19, highlighting a better performance of MESR than SESR (see lines 56-62 of Supplementary Materials). Besides, the uncertainties from the different reanalysis datasets have been evaluated in the Text S1 (see Text S1 in 2-22 of Supplementary Materials). It demonstrates the reliability of the findings using ERA5-Land in this study.

$FD_{RZSM}$ identification, based on soil moisture, is categorized as an agricultural drought. $FD_{SESR}$ is identified by atmospheric evaporation demand and is classified as a meteorological drought. $FD_{MESR}$ identification is an improvement on the $FD_{SESR}$ identification method. It can be thought that $FD_{RZSM}$ and $FD_{SESR}/FD_{MESR}$ are independent, while $FD_{MESR}$ is an advanced version of $FD_{SESR}$ (see lines 85-90 and lines 576-581). The spatial heterogeneity of FD characteristics based on the three indicators has been analyzed, combined with the land use types, Aridity Index (AI), and the ratio of mean annual ET and PET in the revised manuscript (see lines 326-339, lines 363-368, lines 380-388, lines 397-405). Furthermore, the innovation of this study has also been supplemented in the revised manuscript (see Section 4.4 in lines 673-685).

lines 85-90

*"$FD_{SESR}$, identified by anomalous evaporative stress, is classified as a meteorological drought. However, $FD_{RZSM}$, determined by soil moisture, is categorized as an agricultural drought. Although they are interrelated, they identify FDs from different perspectives. Exploring FD characteristics from multiple perspectives plays an important role in the comprehensive understanding and effective response to FD events. Therefore, this study employs RZSM and SESR to further analyze regional FD characteristics and investigate the effects of climate factors and background aridity."*

*"However, the SESR application has several problems. When a FD identified via SESR, both SESR and the change in SESR ($\Delta$SESR) are assumed to follow normal distributions. Therefore, the 50th percentiles of $\Delta$SESR ($\Delta SESR_{50th}$) equal 0, indicating no change in SESR ($\Delta$SESR = 0). A $\Delta$SESR below the 40th percentile of $\Delta$SESR ($\Delta SESR_{40th}$) denotes a decrease in the SESR. However, whether SESR and $\Delta$SESR follow normal distributions remains uncertain. If they do not, $\Delta SESR_{40th}$ may not be less than 0. In the study by Gou et al. (2022), the 36th percentile of $\Delta$SESR ($\Delta SESR_{36th}$) corresponded to an increase in SESR, where $\Delta SESR_{36th}$ was greater than 0. This phenomenon might be because SESR gradually decreases during dry periods and increases during precipitation process. Consequently, $\Delta$SESR can be less than 0 during dry periods and greater than 0 during precipitation periods, potentially leading to underestimation or overestimation of FDs."*

*"In this study, a new method based on the multiples of the mean evaporative stress ratio (MESR) for FD identification was developed to address the aforementioned limitations. Unlike SESR, MESR does not rely on the assumption of a specific probability density function (PDF) for evaporative stress ratio (ESR). Instead, MESR and $\Delta$MESR are fitted by multiple PDFs and converted into percentiles. Variable thresholds are employed for identifying FDs to guarantee that the thresholds of MESR and $\Delta$MESR are less than 0, reflecting a lower level of MESR and a decrease in MESR."*

*"There are three main differences between FD identification via MESR and SESR. First, the ESR value is divided by its mean to construct the MESR series instead of normalizing ESR to create SESR. Regardless of whether ESR is standardized as SESR or MESR, their percentile values are unaffected by linear transformations based on ESR. However, whether ESR follows a normal distribution requires further determination, which makes the rationality of normalizing ESR into SESR debatable. Therefore, dividing ESR by its mean and transforming it into MESR can avoid considering the PDF that ESR follows. Second, the percentiles of MESR and $\Delta$MESR for each pentad are fitted using an optimal PDF rather than an EDF for conversion into percentiles. Because the distributions of MESR and $\Delta$MESR are yet unknown, several PDFs are fitted to select the optimal PDF, ensuring more precise percentiles. Finally, the variable thresholds of MESR and $\Delta$MESR are employed in the process of FD identification. Due to uncertainty regarding whether SESR and $\Delta$SESR follow a normal distribution, and $SESR_{40th}$ and $\Delta SESR_{40th}$ might be greater than 0, variable percentiles of MESR and $\Delta$MESR are used as thresholds for FD identification. The variable thresholds ensures that the threshold is less than 0."*

***"3.1 Shortcomings of $FD_{SESR}$ and rationality of $FD_{MESR}$***

*To graphically illustrate the limitations of the $FD_{SESR}$ method, the distributions of $SESR_{50th}$ and $\Delta SESR_{50th}$ are shown in Fig.3 (a) (b). SESR and $\Delta$SESR percentiles are calculated on a grid by pentads, meaning that $SESR_{50th}$ and $\Delta SESR_{50th}$ vary across grids and pentads, and are temporally and spatially affected. Theoretically, SESR and $\Delta$SESR follow a standard normal distribution, where $SESR_{50th}$ = 0 and $\Delta SESR_{50th}$ = 0. This implies that $SESR_{40th}$ should be below the average level and that $\Delta SESR_{40th}$ should be*

*less than 0, indicating a decreasing SESR. However, the skewed distributions of $SESR_{50th}$ and $\Delta SESR_{50th}$ in the NCP demonstrate that $SESR_{50th}$ and $\Delta SESR_{50th}$ are generally not 0. When $SESR_{50th} > 0$, $SESR_{40th}$ may also exceed 0, indicating that $ESR_{40th}$ exceeds $\overline{ESR}$ and that $SESR_{40th}$ cannot reliably indicate a low evaporative stress value. When $\Delta SESR_{50th} > 0$, $\Delta SESR_{40th}$ may also exceed 0, reflecting an increasing SESR. This could result in underestimation of the evaporative stress value and inaccurate capture of FD events that do not actually occur. When $SESR_{50th} < 0$, $ESR_{40th}$ may be significantly lower than $\overline{ESR}$, indicating a lower evaporative stress value. When $\Delta SESR_{50th} < 0$, $\Delta SESR_{40th}$ is also significantly less than 0, indicating a severely decreasing SESR. This could lead to overestimation of the evaporative stress value and failure to detect FD events that have actually occurred.*

[Figure]

***Figure 3 The distribution of (a) $SESR_{50th}$, (b) $\Delta SESR_{50th}$, the optimal PDF that (c) $MESR_{50th}$ and (d) $\Delta MESR_{50th}$ follow, and distribution of (e) $Pr_1$ and (f) $Pr_2$.***
*Because both MESR and SESR are obtained by the linear transformations of ESR, it can be assumed that MESR also follows a skewed distribution, corresponding to Fig.3 (a) (b). Figure 3 (c) (d) shows the optimal PDFs for MESR and $\Delta MESR$ ($f_1$ and $f_2$). The typical distributions for MESR ($f_1$) and $\Delta MESR$ ($f_2$) are the GEV and logistic distributions. $Pr_1$ and $Pr_2$ are shown in Fig.3 (e) and (f). While some $Pr_1$ and $Pr_2$ values may still deviate from the 50th percentile, there is a notable improvement over the SESR and $\Delta SESR$ distributions in Fig.3 (a) (b). In addition, the variable thresholds used in $FD_{MESR}$ identification ensure that the thresholds related to $Pr_1$ and $Pr_2$ remain less than 0. Therefore, the $FD_{MESR}$ method, where MESR is fitted by several PDFs for FD identification, theoretically offers a more accurate method for identifying FD"*

lines 326-339
*"The frequency of $FD_{RZSM}$ is high in the central and northeastern NCP and low in the southern NCP, with two high-frequency regions in the central and northeastern NCP. The AI values in the central and northeastern NCP are 0.2 ~ 0.3 and less than 0.2, respectively (Fig.S1 (b)), indicating relatively dry conditions. The lack of precipitation and increased evapotranspiration accelerate the decrease in soil moisture, increasing the likelihood of FDs in the central NCP compared with the southern NCP (Gou et al., 2022; Yuan et al., 2023). In addition, the northeastern NCP is predominantly woodland with high ET (Guo et al., 2007) and experiences frequent $FD_{RZSM}$ due to high water demand. In contrast, $FD_{SESR}$ and $FD_{MESR}$ frequencies are both high in the north-central NCP and low in the northeastern and southern NCP. It is inversely correlated with the ratio of annual ET to PET (Fig.S1 (c)), indicating that a region with greater evaporative stress would experience more $FD_{SESR}$ and $FD_{MESR}$. The north-central NCP is primarily cultivated land and is influenced by irrigation. This leads to the significant fluctuations in evaporative stress and results in higher $FD_{SESR}$ and $FD_{MESR}$ frequencies.*

*In contrast, the northeastern NCP, dominated by woodland, is usually not irrigated artificially. Evaporative stress is influenced mostly by climate conditions. Therefore, the northeastern NCP experiences fewer evaporative stress fluctuations and lower $FD_{SESR}$ and $FD_{MESR}$ frequencies (Guo et al., 2007). The southern NCP is characterized by higher temperatures, greater evapotranspiration, and more abundant precipitation as latitude decreases. The balanced hydrothermal conditions in the southern NCP leads to a greater AI and lower FD frequency.”*

[Figure]

**Figure S1 Spatial distribution of (a) the land use in 2010, (b) AI, and (c) the ratio of annual ET and PET in NCP.**

lines 363-368

*“Warmer temperatures may result in longer FD durations (Zhang et al., 2022c). Woodlands take longer to recover from drought than cultivated lands do (Wu et al., 2024). Additionally, human activities might influence FD duration$_{Total}$. Irrigation could significantly alleviate FDs in cultivated land, whereas woodlands in the northeastern NCP are less impacted by human activity and might have a longer FD duration$_{Total}$. The density distributions of duration$_{Total}$ for $FD_{SESR}$ and $FD_{MESR}$ are similar, predominantly ranging between 7 and 9 pentads, whereas that of $FD_{RZSM}$ is lower, ranging between 5 and 9 pentads. This is connected to their identification methods.”*

lines 380-388

*“Overall, the severities of $FD_{RZSM}$ and $FD_{SESR}$ exhibit spatial distributions similar to their duration$_{Total}$ distributions. $FD_{RZSM}$ severity is high in the southern, eastern, and northeastern NCP and low in the central NCP, whereas $FD_{SESR}$ severity is greater in the southern NCP and lower in the northern NCP. Zhang et al. (2021) reported that the spatial distribution patterns of severity and duration$_{Total}$ were similar in the Gan River Basin. Warming exacerbates drought severity by increasing surface evapotranspiration losses and decreasing soil moisture (Yuan et al., 2019; Zhang et al., 2021). High temperatures also the duration of drought in the southern NCP, which might also result in high severity. In contrast, $FD_{MESR}$ severity is high in the northern and central NCP and low in the southern NCP. It is inversely correlated with the ratio of annual ET to PET (Fig.S1 (c)). These findings indicate that $FD_{MESR}$ duration$_{Total}$ and severity are more strongly correlated with evaporative stress than $FD_{SESR}$ is.”*

lines 397-405

*“Lower intensity values indicate more intense FD events. $FD_{RZSM}$ intensity values are low in the southern, northern, and eastern NCP and high in the central NCP. However, $FD_{SESR}$ and $FD_{MESR}$ intensity values are low in the northern NCP and high in the southern NCP. Intensity reflects the rate of decrease in RZSM, SESR, and MESR. For $FD_{RZSM}$, RZSM percentiles decrease more slowly in the west-central NCP. The west-central NCP is located within an arid state with an AI ranging 0.2 ~ 0.4. But it is cultivated land, and irrigation has a significant impact on soil moisture, effectively mitigating the decline in RZSM. Although the AI in the northern NCP is less than 0.4,*

*the woodlands in this region are less impacted by human activities, such as irrigation, which causes RZSM to rapidly decrease. Additionally, high temperatures and great ET in the southern NCP accelerated the RZSM decline. For FD$_{SESR}$ and FD$_{MESR}$, high ET in the southern NCP might not result in a low ESR value or high evaporative stress (Fig.S1). Abundant precipitation and low evaporative stress slow the decline in SESR and MESR."*

lines 477-499

*"**3.4 Typical historical events***

*To evaluate the applicability of the three FD identification methods, two typical drought events occurring in 2017 and 2019 were analyzed (Chen et al., 2024; Xue, 2023; Yao et al., 2022). Xue (2023) identified FD events in the NCP between 1978 and 2020 using soil moisture and reported that a FD$_{RZSM}$ event began in late July 2017, peaked in early August, and terminated by mid-August. Figure 9 (a) shows the spatial evolution of FD$_{RZSM}$, FD$_{SESR}$, and FD$_{MESR}$ from July to August 2017. FD$_{SESR}$ and FD$_{MESR}$ began in the southwestern NCP on July 5th, were alleviated by August 4th, and were followed by sporadic FDs. A FD$_{RZSM}$ event started on July 15th and eased until August 9. After that, the affected area rapidly shrank and ended on August 29th. In late July, the affected area of FD$_{RZSM}$, FD$_{SESR}$, and FD$_{MESR}$ were all large, indicating a severe FD. Furthermore, FD$_{SESR}$ and FD$_{MESR}$ started and developed before FD$_{RZSM}$, indicating that they may precede and influence FD$_{RZSM}$. Therefore, the FD$_{RZSM}$, FD$_{SESR}$, and FD$_{MESR}$ occurred in 2017 in this study align with the findings of Xue (2023).*

[Figure]

***Figure 9 The spatiotemporal evolution process of FD events in (a) 2017 and (b) 2019.***

[Figure]

**Figure 9 (Continued).**

*Yao et al. (2022) analyzed the 2019 FD event and discovered that FD$_{RZSM}$ developed rapidly from April 30th to June 9th, during which the RZSM percentiles decreased sharply from 86% to 25%. Afterward, the RZSM percentile decreased once again, FD$_{RZSM}$ severity peaked in July, and recovered in August. Figure 9 (b) shows that FD$_{RZSM}$ started on April 26th and briefly recovered on June 5th. The FD$_{RZSM}$ worsened again on June 20th, reached its peak affected area from late June to early July, and gradually recovered and terminated in August. FD$_{SESR}$ and FD$_{MESR}$ exhibited evolution similar to that of FD$_{RZSM}$. It began on April 26th, recovered on May 31st, worsened again on June 15th, eased on July 10th, and ended on July 30th. The evolution is consistent with that reported by Yao et al. (2022). Therefore, FD identification via RZSM, SESR, and MESR in this study might be in good agreement with the actual FD events, and the applicability of FD identification methods used in this study is validated."*

lines 576-581

*"Agricultural FDs (FD$_{RZSM}$) are less frequent than meteorological FDs (FD$_{SESR}$ and FD$_{MESR}$). The FD$_{RZSM}$ identification is based on soil moisture from an agricultural perspective, whereas FD$_{SESR}$ and FD$_{MESR}$ focus on atmospheric evaporation demand from a meteorological perspective. FD$_{RZSM}$ and FD$_{SESR}$/FD$_{MESR}$ are independent, whereas FD$_{MESR}$ is an advanced version of FD$_{SESR}$. The higher FD$_{SESR}$ and FD$_{MESR}$ frequencies than FD$_{RZSM}$ frequency is related to the fact that meteorological FDs are directly determined by atmospheric conditions and is more sensitive to FD occurrence and development. In contrast, agricultural FDs are dependent on soil moisture and influenced by crop regulatory effects and human activities (Meng et al., 2024)."*

lines 607-653

*"**4.2 Attribution analysis of frequency differences between FD$_{SESR}$ and FD$_{MESR}$***
*Compared with FD$_{SESR}$, FD$_{MESR}$ frequency shows spatial heterogeneity, which is related to the frequency distributions of SESR$_{50th}$ and ΔSESR$_{50th}$. Figure S11 illustrates the spatial distributions of the mean SESR$_{50th}$ and ΔSESR$_{50th}$ during FD$_{SESR}$ pentads. Except for the southern NCP, where SESR$_{50th}$ is less than 0, SESR$_{50th}$ in the northern and central NCP is predominantly greater than 0. This may result in SESR$_{40th}$ failing*

*to represent the real low evaporative stress value and an underestimation of the evaporative stress value. In addition, ΔSESR50th is greater than 0 in the NCP, which may be accompanied by increasing SESR and inaccurate capture of FDSESR events that do not occur, potentially causing overestimation in the NCP (see Sect.3.2). Theoretically, SESR40th and ΔSESR40th in the NCP should be less than 0, representing a low evaporative stress value and a decreasing ESR. The distributions of average SESR40th and ΔSESR40th in Fig.S11 (c) (d) indicate that both are less than 0, except in the northern NCP, where SESR40th exceeds 0. This further shows that unobserved FDSESR events might be captured in the northern NCP. The histograms of SESR50th, ΔSESR50th, SESR40th, and ΔSESR40th during FDSESR events are shown in Fig.S12. SESR50th and ΔSESR50th values are mostly greater than 0, particularly ΔSESR50th, corresponding to Fig.S11 (a) (b). However, most SESR40th and ΔSESR40th are less than 0. Notably, approximately 35% of SESR40th and ΔSESR40th are greater than 0, supporting the likelihood of FDSESR overestimation in the NCP. Notably, the difference between Fig.3 (a) (b) and Fig.S12 (a) (b) is that Fig.S12 (a) (b) shows the SESR50th and ΔSESR50th values during FDSESR occurrence, whereas Fig.3 (a) (b) shows those for all pentads.*

[Figure]

*Figure S11 Distribution of the mean (a) SESR50th, (b) ΔSESR50th, (c) SESR40th, and (d) ΔSESR40th on FDSESR pentads.*

[Figure]

*Figure S12 Histogram of (a) SESR50th, (b) ΔSESR50th, (c) SESR40th, and (d) ΔSESR40th on FDSESR pentads.*

*The FD identification based on SESR and MESR differs in three aspects. First, ESR is divided by its mean to construct MESR rather than being normalized into SESR. Second, MESR is fitted using various PDFs instead of EDF. Third, variable thresholds are utilized in FDMESR identification. Both MESR and SESR are linearly transformed from ESR to facilitate a comparison of FD identification results between different regions.*

*Converting ESR to MESR rather than SESR enhances the rationality of the ESR standardization process. However, the linear transformation of ESR into SESR or MESR does not affect its corresponding percentiles. Therefore, the differences between $FD_{SESR}$ and $FD_{MESR}$ identification results can be attributed to two factors: PDF fitting and variable thresholds.*

*To quantify the contributions of PDF fitting and variable thresholds to the differences between $FD_{SESR}$ and $FD_{MESR}$, a modified version of $FD_{MESR}$, termed $FD_{MESR-invariable}$ was created. The fixed thresholds from the $FD_{SESR}$ method instead of variable thresholds were applied, i.e., MESRonset1 = 40, MESRonset2 = 20, $\Delta$MESRonset = 40, and MaxMESRchange = 25. Figure S13 (a) ~ (c) displays the frequency of $FD_{MESR-invariable}$ and the differences between $FD_{MESR-invariable}$ frequency and $FD_{SESR}$ and $FD_{MESR}$ frequencies. Figure S13 (d) (e) shows the contributions of PDF fitting and variable thresholds to the differences between $FD_{SESR}$ and $FD_{MESR}$ frequencies, and Fig.S13 (f) illustrates their relative contributions.*

[Figure]

**Figure S13 Frequency of $FD_{MESR-invariable}$ and its difference with $FD_{MESR}$ and $FD_{SESR}$.** *$FD_{MESR-invariable}$ and $FD_{MESR}$ frequencies exhibit similar spatial distributions, with higher frequencies in the north-central NCP and lower frequencies in the southern NCP. In addition, $FD_{MESR-invariable}$ has a lower frequency than both $FD_{SESR}$ and $FD_{MESR}$ do, indicating that PDF fitting decreases FD frequency, whereas variable thresholds increase it. According to Fig.S13 (d) (e), PDF fitting has a negative effect on the difference between $FD_{SESR}$ and $FD_{MESR}$ frequencies in the northeastern and west-central NCP, whereas variable thresholds have a positive effect. However, it is opposite in the other NCP regions. As shown in Fig.S13 (f), the relative contribution of variable thresholds to PDF fitting mostly ranges between -2 and -1 in the northeastern and west-central NCP but between -1 and 0 in other regions. Therefore, the absolute contribution of PDFs fitting is greater than 1 in the northeastern and west-central NCP but less than 1 in other regions. Considering the negative contribution of PDF fitting in the NCP, it*

*can be assumed that the contribution of PDF fitting is less than that of variable thresholds in the northeastern and west-central NCP but greater in other regions. Therefore, the frequency difference between FD$_{SESR}$ and FD$_{MESR}$ frequencies is driven mainly by variable thresholds in the northeastern and west-central NCP, whereas PDF fitting plays a greater role in other regions.*

*FD$_{SESR}$ frequency is lower than FD$_{MESR}$ frequency in the northeastern and west-central NCP but higher in other regions. Although FD$_{SESR}$ frequency is overestimated in the NCP, particularly in the northern NCP, PDF fitting decreases FD frequency in the FD$_{MESR}$. However, in the northeastern and west-central NCP, the increase in FD frequency due to variable thresholds outweighs the decreases due to PDF fitting, resulting in a higher FD$_{MESR}$ frequency than FD$_{SESR}$ frequency. In other regions, the decrease in FD frequency from PDF fitting surpasses the increase from variable thresholds, leading to a lower FD$_{MESR}$ frequency than FD$_{SESR}$ frequency."*

lines 2-22 of Supplementary Materials
*"**Text S1 Uncertainties from the reanalysis datasets***

*To assess data-related uncertainties, the soil moisture, ET, and PET obtained from two additional reanalysis datasets, the GLEAM and GLDAS 2 datasets, were utilized to identify FD$_{RZSM}$, FD$_{SESR}$, and FD$_{MESR}$. Because the RZSM, SESR, and MESR are the basis for FD identification, their pentad percentiles were determined. Figure S14 shows the Taylor diagrams comparing the pentad RZSM, SESR, and MESR percentile series from 1981 to 2022 for the ERA5-Land, GLEAM, and GLDAS 2 datasets, with the ERA5-Land dataset as the reference. The pentad RZSM, SESR, and MESR percentiles from the GLEAM and GLDAS 2 datasets are highly consistent. Their correlation coefficients are approximately 0.7, centered root mean square differences are approximately 0.8, and standard deviations are close to 1. Therefore, the pentad percentiles of RZSM, SESR, and MESR of the ERA5-Land dataset are consistent with those of the GLEAM and GLDAS 2 datasets. Figure S15 displays the spatial distributions of correlations for the pentad RZSM, SESR, and MESR percentiles between the ERA5-Land dataset and the GLEAM and GLDAS 2 datasets. As shown in Fig.S15 (a) and (d), the RZSM percentile correlation between the ERA5-Land and the GLEAM is comparable to that between the ERA5-Land and the GLDAS 2, primarily exceeding 0.6. SESR percentile correlation between the ERA5-Land and the GLEAM is similar to MESR percentile correlation, which mostly exceeds 0.5. The correlations of SESR and MESR percentiles between the ERA5-Land and the GLDAS 2 are primarily between 0.4 and 0.7, with a comparable spatial distribution pattern. The greater correlation between the ERA5-Land and the GLEAM than between the ERA5-Land and the GLDAS 2 might be due to the coarse spatial resolution of the GLDAS 2.*

[Figure]

***Figure S14 Taylor diagram for the pentad (a) RZSM, (b) SESR, and (c) MESR percentiles based on ERA5-Land, GLEAM, and GLDAS 2 datasets.***

[Figure]

**Figure S15 Spatial distribution of the Pearson correlation of the pentad RZSM, SESR, and MESR percentiles between ERA5-Land and (a) ~ (c) GLEAM and (d) ~ (f) GLDAS 2 datasets.**

[Figure]

**Figure S5 Histogram of FD characteristics identified by RZSM, SESR, and MESR based on ERA5-Land.**

*In addition, FD characteristics identified using the ERA5-Land, GLEAM, and GLDAS 2 datasets are displayed in Figs.S5, S16 and S17. The distributions of $FD_{RZSM}$, $FD_{SESR}$, and $FD_{MESR}$ characteristics are consistent across the datasets, except for $FD_{MESR}$ intensity based on the GLDAS 2. Moreover, the proportions of various FD grades determined by intensity from diverse datasets also demonstrate strong agreement, as shown in Fig.S6. The similarities in the pentad RZSM, SESR, and MESR percentiles from various datasets and FD characteristics across datasets effectively demonstrate the reliability of our findings."*

[Figure]

**Figure S16 Same as Figure S5, but based on GLEAM.**

[Figure]

**Figure S17 Same as Figure S5, but based on GLDAS 2.**

[Figure]

**Figure S6 Proportion of FD1 ~ 4 for FD$_{RZSM}$, FD$_{SESR}$, and FD$_{MESR}$ based on (a) ERA5-Land, (b) GLEAM, and (c) GLDAS 2 datasets.**

*"**Text S2 Explanatory ability between different FD types***

*Given the influence of climate control on FD occurrence (Mukherjee and Mishra, 2022), there might be relationships between different FD types. Therefore, the coefficient of*

*determination ($R^2$) was used to quantify the relationships among $FD_{RZSM}$, $FD_{SESR}$, and $FD_{MESR}$. $R^2$ represents the capacity of linear regression to explain the variance in the dependent variable based on the independent variables (Mukherjee and Mishra, 2022). In particular, the relationship between $FD_{RZSM}$ and $FD_{SESR}$, denoted as "RZSM ~ SESR", is represented by the $R^2$ derived from the linear regression between RZSM percentile (dependent variable) and SESR percentile (independent variable) during $FD_{RZSM}$ pentads. Moreover, other relationships, such as "RZSM ~ MESR", "SESR ~ RZSM", "MESR ~ RZSM", "SESR ~ MESR", and "MESR ~ SESR", were determined, as shown in the first two columns of Fig.S18.*

[Figure]

**Figure S18 Spatial distribution of the $R^2$ determined by (a) "RZSM ~ SESR", (b) "RZSM ~ MESR", (e) "SESR ~ RZSM", (f) "MESR ~ RZSM", (i) "SESR ~ MESR", and (j) "MESR ~ SESR", as well as the differences of $R^2$ (c) between "RZSM ~ SESR" and "RZSM ~ MESR", (g) between "SESR ~ RZSM" and "MESR ~ RZSM", and (k) between "SESR ~ MESR" and "MESR ~ SESR". The boxplots in the (d), (h), and (l) illustrate the $R^2$ in the "RZSM ~ SESR" and "RZSM ~ MESR", "SESR ~ RZSM" and "MESR ~ RZSM", and "SESR ~ MESR" and "MESR ~ SESR" over different AI values.**

*In Fig.S18 (a) and (b), both "RZSM ~ SESR" and "RZSM ~ MESR" explain more than 40% of the variance in RZSM percentile in the central NCP but less than 30% in other regions. In Fig.S18 (e) and (f), the southern NCP shows higher $R^2$ values for "SESR ~ RZSM" and "MESR ~ RZSM" (mostly approximately 15% ~ 25%) than the northern NCP does (less than 15%). However, "SESR ~ MESR" explains more than 90% of the variance in SESR, and "MESR ~ SESR" explains more than 90% of the variance in MESR, as shown in Fig.S18 (i) and (j). Overall, the explanatory ability of these relationships, ranked from highest to lowest, is "SESR ~ MESR" and "MESR ~ SESR" > "RZSM ~ SESR" and "RZSM ~ MESR" > "SESR ~ RZSM" and "MESR ~ RZSM". Because SESR and MESR are both derived from the linear transformation of ESR, they exhibit strong mutual explanatory ability. The relationships between MESR and RZSM ("RZSM ~ MESR" and "MESR ~ RZSM") are quite comparable to those between SESR and RZSM ("RZSM ~ SESR" and "SESR ~ RZSM"), highlighting the reliability of FD identification using MESR. The differences in Fig.S18 (c), (g), and (k) further demonstrate the similarities between SESR and MESR. However, considering the propagation from meteorological to agricultural drought, "RZSM ~ MESR" has a slightly greater explanatory ability for RZSM than "RZSM ~ SESR" in Fig.S18 (c), illustrating that MESR performs better than SESR in explaining RZSM.*

*The spatial distributions of $R^2$ also reveal sensitivity to the AI, as shown in Fig.S18 (d),*

*(h), and (l). For "RZSM ~ SESR" and "RZSM ~ MESR", the explanatory ability increases with increasing AI in the region where AI < 0.3 but decreases in the region where AI > 0.3. The region with AIs between 0.2 and 0.3 has the highest explanatory ability for RZSM percentile (approximately 60%). Overall, the RZSM percentiles could be better explained by the SESR and MESR percentiles in drier regions, except where AI < 0.2, which might be related to RZSM being greater initially in wetter regions with extended memory (Mukherjee and Mishra, 2022). For "SESR ~ RZSM" and "MESR ~ RZSM", the explanatory ability is less than 20%, and is obviously greater in the region with AI > 0.2 than in the region with AI < 0.2. SESR and MESR percentiles could be better explained by RZSM percentile in wetter regions with less evaporative stress and greater evaporation. Meteorological drought ($FD_{SESR}$ and $FD_{MESR}$) might lead to agricultural drought ($FD_{RZSM}$), resulting in lower $R^2$ for "SESR ~ RZSM" and "MESR ~ RZSM" compared to "RZSM ~ SESR" and "RZSM ~ MESR". For "SESR ~ MESR" and "MESR ~ SESR", the explanatory ability exceeds 90% and generally increases with increasing AI overall."*

lines 56-62 of Supplementary Materials
*"To quantify which indicator, SESR or MESR, can better identify FDs in the NCP, the relationships between the two indicators and the pentad SPI series were measured. The maximal information coefficient (MIC) can measure both linear and nonlinear relationships between two series (Cao et al., 2021). Therefore, the MICs between the pentad SPI series and SESR or MESR percentile series (denoted as $MIC_{SPI\&SESR}$ and $MIC_{SPI\&MESR}$) were calculated grid by grid, and their differences are shown in Fig.S19. The difference between $MIC_{SPI\&SESR}$ and $MIC_{SPI\&MESR}$ is predominantly greater than 0, demonstrating a stronger correlation between SPI and MESR than between SPI and SESR. This suggests that the performance of MESR is better than that of SESR in the meteorological drought identification in the NCP."*

[Figure]

**Figure S19 Difference between $MIC_{SPI\&SESR}$ and $MIC_{SPI\&MESR}$.**